# When Language Models Lose Their Mind: The Consequences of Brain Misalignment

**Gabriele Merlin**
MPI-SWS
Saarbrücken, Germany
gmerlin@mpi-sws.org

**Mariya Toneva**
MPI-SWS
Saarbrücken, Germany
mtoneva@mpi-sws.org

## Abstract

While brain-aligned large language models (LLMs) have garnered attention for their potential as cognitive models and for potential for enhanced safety and trustworthiness in AI, the role of this brain alignment for linguistic competence remains uncertain. In this work, we investigate the functional implications of brain alignment by introducing brain-misaligned models–LLMs intentionally trained to predict brain activity poorly while maintaining high language modeling performance. We evaluate these models on over 200 downstream tasks encompassing diverse linguistic domains, including semantics, syntax, discourse, reasoning, and morphology. By comparing brain-misaligned models with well-matched brain-aligned counterparts, we isolate the specific impact of brain alignment on language understanding. Our experiments reveal that brain misalignment substantially impairs downstream performance, highlighting the critical role of brain alignment in achieving robust linguistic competence. These findings underscore the importance of brain alignment in LLMs and offer novel insights into the relationship between neural representations and linguistic processing.

## 1 Introduction

A growing body of work studies the intriguing parallels between pretrained large language models (LLMs) and the human brain, demonstrating a substantial degree of alignment between brain activity patterns and LLM activations when humans and LLMs are presented with the same linguistic input (Toneva & Wehbe, 2019; Caucheteux & King, 2020; Schrimpf et al., 2021; Goldstein et al., 2022; Aw & Toneva, 2023; Merlin & Toneva, 2024; Karamolegkou et al., 2023). This existing brain-LLM alignment has excited both cognitive scientists and AI researchers. From a cognitive perspective, brain-aligned LLMs can serve as model organisms for studying natural language processing in the human brain, offering insights into mechanisms that may underlie human-like linguistic behavior and representation (Toneva, 2021). From an AI perspective, researchers posit that brain-aligned LLMs may be safer and more trustworthy (Mineault et al., 2024). Relatedly, a recent study demonstrated the first substantial downstream benefits of improving brain alignment of a speech language model, by showing that brain-tuning a model significantly improves its performance on downstream semantic tasks (Moussa et al., 2025; Vattikonda et al., 2025).

Despite this promise of brain-LLM alignment, its necessity for model performance remains an open question. It is unclear whether alignment with the human brain is inherently required for LLMs to perform well on linguistic tasks, or whether the relationship between brain alignment and model behavior is more nuanced. To address this gap, it is essential to understand not only the presence of alignment but also its functional implications.

In this work, we take a direct approach to investigate the effect of brain alignment on LLM performance. We introduce brain-misaligned models–language models specifically trained to predict brain activity poorly while maintaining robust language modeling performance on the same linguistic inputs. We evaluate these models across more than 200 downstream tasks spanning a broad spectrum of linguistic capabilities, including semantics, syntax, discourse, reasoning, and morphology. By comparing brain-misaligned models with well-matched models that differ primarily in their ability to predict brain activity rather than their language modeling proficiency, we isolate the impact of

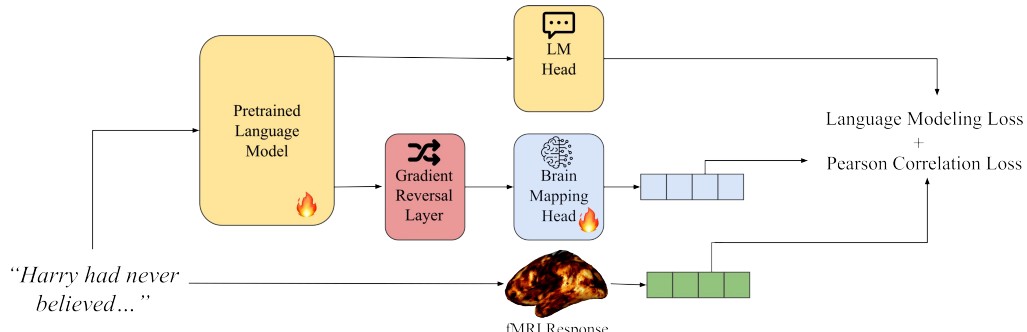

Figure 1: A schematic of the proposed approach. Our method is based on fine-tuning a pretrained language model with two simultaneous objectives: maintaining its language modeling ability while reducing its alignment with brain recordings. Language modeling performance is preserved by continuing training on a fine-tuning dataset using the standard language modeling objective. Brain alignment is reduced by introducing a second prediction head and a gradient reversal layer, which encourages the model to produce representations that are uninformative about the corresponding brain activity.

brain alignment on downstream linguistic performance. Our results reveal that brain-misalignment significantly impairs the ability of LLMs to perform linguistic tasks. These findings suggest that alignment with the human brain is crucial for LLMs to achieve strong linguistic performance, shedding light on the functional relevance of brain alignment in modern language models.

Our main contributions can be summarized as follows:

1. We develop brain-misaligned models that allow us to investigate the effect of brain alignment on the linguistic competence of language models.

2. We evaluate the effect of brain misalignment on a comprehensive set of linguistic tasks, comprising more than 200 datasets. These tasks are designed to assess various linguistic subfields (syntax, semantics, discourse, reasoning, and morphology) and linguistic phenomena (e.g., part of speech, protoroles, coreference resolution).

3. Via comparisons with well-matched controls, we show that brain misalignment significantly decreases linguistic competence. This suggests that brain alignment is necessary to maintain linguistic competence in language models.

4. We find that the competence drop is especially pronounced in semantic and syntactic tasks, demonstrating the importance of brain alignment for language models.

5. To further validate our findings, we also finetune a model using brain recordings, showing that the model improves in every linguistic subfield with respect to other finetuned models, and is also better than pretrained models, in particular for semantics and syntax tasks.

## 2 RELATED WORKS

A growing body of research investigates the alignment between pretrained language models and human brain activity during language comprehension (Wehbe et al., 2014b; Jain & Huth, 2018; Toneva & Wehbe, 2019; Abdou et al., 2021; Schrimpf et al., 2021; Hosseini et al., 2024). Other studies have focused on understanding the factors that drive this alignment, identifying model characteristics or representational properties that correlate with neural responses (Goldstein et al., 2022; Toneva et al., 2022a; Oota et al., 2024a;b; Caucheteux et al., 2021; Reddy & Wehbe, 2021; Toneva et al., 2022b; Kauf et al., 2023; Gauthier & Levy, 2019; Aw & Toneva, 2023; Merlin & Toneva, 2024). Additionally, previous work have started to use brain data for finetuning language models (Schwartz et al., 2019) showing that is possible to improve downstream performance of pretrained language model (Negi et al., 2025). Our work extends these findings by investigating whether this alignment is not only observed but also functionally relevant for language processing, specifically, whether brain alignment is necessary for maintaining linguistic competence in language models.

In fact, a substantial body of work has focused on evaluating the linguistic competencies of language models. These studies aim to systematically assess the extent to which models capture various linguistic phenomena, including syntax, semantics, morphology, and discourse-level reasoning (Amouyal et al., 2024; Blevins et al., 2023). Benchmarks such as BLiMP (Warstadt et al., 2020), GLUE (Wang et al., 2018), SuperGLUE (Wang et al., 2019), and more recently Holmes (Waldis et al., 2024) have been used to evaluate models' understanding of language. Our study contributes to this line of research by examining how these linguistic competencies are affected when the alignment between language model representations and brain activity is manipulated.

Additionally, a growing line of work in Causal NLP aims to uncover causal relationships between model components, training signals, or representations and downstream performance (Feder et al., 2021; 2022; Liu et al., 2025; Ortu et al., 2024). These studies design interventions or counterfactual setups to test whether certain features are causally implicated in model predictions or behaviors. Our approach is aligned with those works. We intervene on brain alignment, training models to preserve or disrupt alignment, and estimate its causal role in supporting linguistic competence.

## 3 METHODOLOGY

### 3.1 PRETRAINED MODELS

We use BERT-based (Devlin et al., 2019), GPT2-based (Radford et al., 2019) and Llama-based (Liu et al., 2024) language models. In particular, we focus on the `bert-base-cased`, `gpt-small` and `meta-llama/Llama-3.2-1B` provided by Hugging Face (Wolf et al., 2020). BERT, GPT2 and Llama have achieved strong performance on various NLP tasks, such as question answering and sentence classification. Moreover, they have been extensively studied in prior work on brain alignment (Toneva & Wehbe, 2019; Caucheteux et al., 2021; Oota et al., 2024b).

### 3.2 FMRI DATA

We use two publicly available fMRI datasets to measure the brain alignment of language model representations. The data included in the first dataset, provided by Wehbe et al., 2014a, were collected from eight participants as they read Chapter 9 of Harry Potter and the Sorcerer's Stone (Rowling et al., 1998) word by word. The chapter was divided into four runs of similar length, each separated by a short break. Each word was presented for 0.5 seconds, and one fMRI image (TR) was acquired every 2 seconds, resulting in 1211 brain images per participant. The fMRI data in the second dataset, made publicly available by Deniz et al., 2019, consist of recordings from six participants who read and listened to the same 11 stories from The Moth Radio Hour. For each modality, the dataset includes 4028 fMRI images. During reading, each word was presented for exactly the same duration as in the audio recording. In our analysis, we used only the reading data. These datasets are among the largest publicly available collections in terms of the amount of data per participant, which is crucial for obtaining accurate estimates of brain alignment.

### 3.3 CONTROLLING BRAIN ALIGNMENT

To investigate the effect of brain alignment of a language model on its downstream linguistic competence, we develop three models: the Brain Misaligned model, the Brain Preserving model and the Brain Tuned model. The Brain Misaligned model is trained to reduce alignment with brain recordings, while the Brain Preserving model serves as a comparison baseline that preserves brain alignment while controlling for possible confounding factors. We also designed a Brain Tuned model that is trained to improve alignment with brain recordings. This model serves to further validate our analysis.

### 3.3.1 BRAIN MISALIGNED MODEL

To evaluate the influence of brain-related information, which is an abstract concept for which no clear counterfactual input exists, we must develop methods that allow us to remove such information directly from the language model representations. In this study, we address this challenge by designing an intervention to language models that aims to remove brain-related information from

their representations, without the need to generate counterfactual inputs. This enables us to investigate the necessity of brain alignment for natural language processing abilities.

Our approach is based on adversarial fine-tuning (Ganin et al., 2016) of language models, using a prediction head (*brain mapping head* in Figure 1) and a gradient reversal layer to remove the targeted capacity, i.e. brain prediction, while simultaneously fine-tuning a second head to preserve the language modeling performance.

The model is finetuned using the stimuli from the Harry Potter fMRI dataset (Wehbe et al., 2014a) or from the Moth Radio Hour fMRI dataset for the language modeling loss, and the corresponding fMRI recordings for the loss of the brain mapping head. For training, we select only voxels with an estimated noise ceiling $> 0.05$ (see Appendix C for details) belonging to regions of the brain known to process language (Fedorenko et al., 2010; Fedorenko & Thompson-Schill, 2014; Binder et al., 2009; Oota et al., 2024a) and used by previous works to investigate brain alignment of language models. Additional details on the prediction of brain recordings are reported in Appendix B. The total loss is defined as:

$$\mathcal{L} = \omega_{lm} * \mathcal{L}_{lm} + \omega_{ba} * \mathcal{L}_{ba}$$

where $\mathcal{L}_{lm}$ is the language modeling loss, $\mathcal{L}_{ba}$ is the brain-alignment loss, and $\omega_{lm}$ and $\omega_{ba}$ are weighting factors to balance the two objectives. The language modeling loss $\mathcal{L}_{lm}$ corresponds to the standard cross-entropy loss used during language model pretraining, while the brain-alignment loss $\mathcal{L}_{ba}$ is defined as the mean negative squared Pearson correlation between the predicted voxels in each batch and the ground truth voxel values. $\omega_{lm}$ is fixed at 0.1, a value chosen based on the relative magnitude of the losses prior to fine-tuning (see Section 3.3.4 for details).

### 3.3.2 BRAIN PRESERVING MODEL

Similarly, we designed a control condition to account for potential confounding factors and to serve as a comparison for the Brain Misaligned model. We finetune this model using the same procedure as for the Brain Misaligned model, but during training we permute the order of the fMRI images to disrupt the correspondence between stimuli and brain activity.

By using permuted fMRI images, our method also accounts for the effects of the adversarial removal itself, which can influence the model's representations. This controls for potential confounders such as the effect of fine-tuning on language modeling and the effect of adversarial fine-tuning. The only difference between conditions remains the correspondence between stimuli and fMRI images.

### 3.3.3 BRAIN TUNED MODEL

To complement our analysis, we designed a Brain Tuned model. This model was finetuned using the same procedure as the Brain Misaligned model (i.e. a language modeling head and a brain mapping head), but we removed the gradient reversal layer and use as loss function $\mathcal{L}_{ba}$ the mean negative Pearson correlation. This procedure actively encourages the model to increase its alignment with brain recordings while maintaining language modeling performance. This model serves as a validation tool to test the complementary hypothesis: if decreasing alignment impairs competence, does increasing it lead to performance gains?

### 3.3.4 MODEL SELECTION AND TRAINING

To train the models, we use training samples consisting of sequences of words corresponding to 5 TRs. The stimulus text is divided into four consecutive sections to enable cross-validation.

For training of BERT-based, GPT2-based and Llama-based Misaligned, Brain Preserving and Brain Tuned models we train applying LoRA (Hu et al., 2022) to the parameters. We train for 5 epochs with a batch size of 16, and AdamW as optimizer. The language modeling loss weight $\omega_{lm} = 0.1$ and $\omega_{ba} = 10$.

**Conditions for a successful comparison between models.** The comparison is considered successful when the Brain Misaligned and Brain Preserving models achieve similar performance on the language modeling objective (tested using Wilcoxon signed-rank test, $p < 0.05$), while the Brain Misaligned model shows a significantly lower ability to align with brain recordings.

### 3.4 EVALUATION

To evaluate the models, we use three types of tasks: language modeling, brain alignment, and downstream linguistic tasks. Both language modeling and brain alignment are evaluated using the same text, which corresponds to the fMRI stimulus, and is held-out during training. We assess these two tasks using overlapping sequences of words belonging to 5 TRs, following the approach of previous work (Merlin & Toneva, 2024).

**Language modeling.** For language modeling we follow the best practice for evaluation of BERT-based, GPT2-based and Llama-based models. For each test example, we measure the average cross entropy across the randomly masked tokens (15% of total number of tokens, see Devlin et al. (2018) for details) for BERT-based models, for GPT2-based and Llama-based models the cross entropy over all tokens (see Radford et al. (2019) for details).

**Brain alignment.** We measure the brain alignment between BERT, GPT2 and Llama representations and fMRI recordings using a linear prediction head on top of the last transformer block. This prediction head is trained to output brain activity values from the model's representations and is widely used in previous work to assess how well language models can predict brain signals (Jain & Huth, 2018; Toneva & Wehbe, 2019; Schrimpf et al., 2021). We train this linear function, regularized with a ridge penalty, using cross-validation and evaluate its performance on held-out data. The ridge parameter is selected via nested cross-validation. Consequently, for each participant, we train one model for each held-out run (see Section 3.2), then aggregate the predictions to compute brain alignment. Further details on the prediction head are provided in Appendix B. Brain alignment is quantified using Pearson correlation, computed between the predictions on held-out data and the corresponding ground truth values. Specifically, for a model $q$ and voxel $v_j$ with corresponding held-out fMRI values $y_j$, brain alignment is computed as:

$$\texttt{brain alignment}(q, v_j) = \texttt{corr}(\hat{y_j}, y_j),$$

where $\hat{y_j} = q(X)W_{q,j}$, $X$ is the input text sample to model $q$, and $W_{q,j}$ are the learned prediction weights for voxel $v_j$.

**Linguistic competence.** To investigate the linguistic competence of language models, we use more than 200 datasets, designed to evaluate linguistic competence in language models via classifier-based probing (Waldis et al., 2024). The benchmark covers datasets spanning various linguistic phenomena and subfields, including syntax, morphology, semantics, reasoning, and discourse, examples of tasks are reported in Appendix Table 1. Details about the benchmark and the included datasets are provided in Appendix A. For each task, each model is evaluated using 6 seeds, which influence the probe initialization and the ordering of data during training and evaluation.

To determine whether one model outperforms the other, we not only compare the average evaluation metric (see Waldis et al. (2024) for details), but also assess whether the difference is statistically significant using a two-sample t-test. We assign a "win" to a model only for datasets where the difference reaches statistical significance. For each dataset and model pair, we thus obtain a binary "win" matrix indicating whether one model significantly outperforms the other (1) or not (0). Since each subject has a pair of models corresponding to different held-out runs during training, we average the resulting win matrices across runs, yielding a win score for each participant, dataset, and model. The win score quantifies how consistently one model outperforms the other across different held-out runs.

## 4 RESULTS

### 4.1 EFFECTS ON BRAIN ALIGNMENT

Figure 2A–D shows brain alignment of the BERT-based Brain Misaligned and Brain Preserving models on the Harry Potter dataset, as well as a contrast between the two, for a representative participant. Specifically, Figures 2A and 2B show the Pearson correlation between the predicted voxel values and the ground truth for the Brain Preserving model and the Brain Misaligned model, respectively. Figure 2C shows the contrast between the two models, i.e., the difference in Pear-

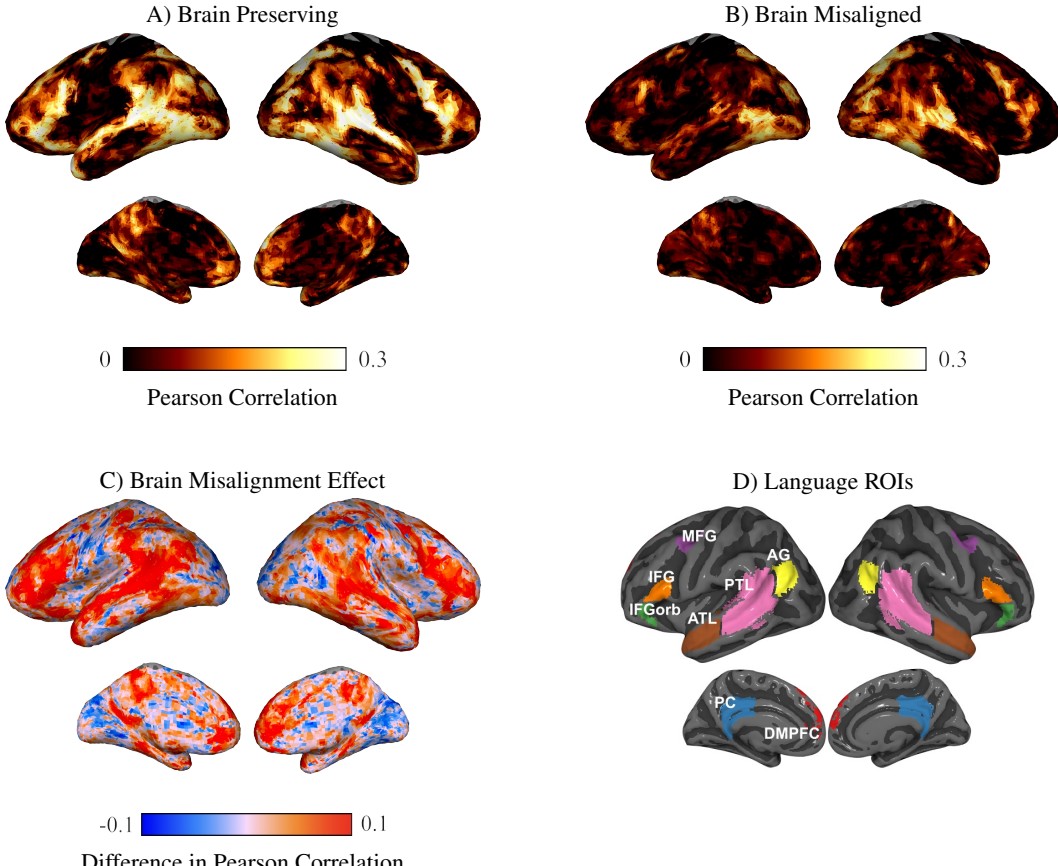

Figure 2: Brain alignment of the BERT-based Brain Preserving (A) and Brain Misaligned (B) models for one participant on the Harry Potter dataset (see Appendix D for all participants), and the difference between the two (C). The Brain Misaligned model exhibits substantially weaker alignment, particularly in language regions (C, D).

son correlation for each voxel. Results for the remaining participants, other language models and datasets are consistent and reported in Appendix D, E, F, G, H, I.

**Brain Preserving Model.** In Figure 2A, we observe that the Brain Preserving model aligns well with brain activity across the whole brain, and in particular within language-related regions (visualized in Figure 2D), as identified by previous work (Fedorenko et al., 2010; Fedorenko & Thompson-Schill, 2014; Binder et al., 2009). We quantify the alignment of the Brain Preserving model in Appendix D. Since these models are explicitly finetuned to preserve brain alignment, this result is consistent with prior studies showing that language models exhibit alignment with fMRI signals (Toneva & Wehbe, 2019; Schrimpf et al., 2021; Goldstein et al., 2022; Oota et al., 2024b; Merlin & Toneva, 2024).

**Brain Misaligned Model.** In Figure 2B, we observe that the Brain Misaligned model does not align well with brain activity across the whole brain. The Pearson correlation values are particularly low in several brain areas. We quantify the alignment of brain misaligned model in Appendix D.

To show the effect of brain misalignment, in Figure 2C we show the contrast between the Pearson correlation values of the Brain Preserving model and the Brain Misaligned model. As expected, the difference in Pearson correlation is especially high in language-related regions (visualized in Figure 2D). This confirms that our approach is effective at removing brain-relevant information in particular in language-related areas.

We assess whether the average brain correlation (computed across voxels in language regions with an estimated noise ceiling $> 0.05$, see Appendix C) significantly differs between the two models using a t-test. For example, for BERT trained on Harry Potter dataset we find that for six participants, there is a significant drop in brain alignment, while no significant difference in language modeling ability is observed between the two models. Only the models corresponding to these participants are included in the comparison of linguistic competence.

## 4.2 Effects on Linguistic Competence

Figures 3A, 3B, and 4 show the performance on linguistic competence averaged across models and dataset combinations (BERT-Harry, BERT-Moth, GPT2-Harry, GPT2-Moth, Llama-Harry, Llama-Moth). Specifically, Figure 3A illustrates the overall effect on linguistic competence, considering all tasks. Figure 3B presents the results by linguistic subfields, while Figure 4 focuses on specific linguistic phenomena. Specific results for each combinations of models and dataset are reported in Appendix E, F, G, H, I.

**Effects on the Overall Linguistic Competence.** Figure 3A shows the average win rate in the linguistic competence benchmark for the Brain Misaligned models and the Brain Preserving models. We observe a significant difference between the two conditions. These results indicate that removing brain alignment leads to lower performance on downstream linguistic tasks, suggesting that brain alignment is necessary to preserve linguistic competence. Statistical tests on each individual model and dataset combinations reveal a marked difference in performance between the Brain Misaligned and Brain Preserving on the overall linguistic competence. For the BERT-based and Llama-based models the difference is significant, although for GPT2-based models on the Harry Potter dataset the difference does not reach conventional statistical significance (p-value $= 0.055$), the trend mirrors the effect observed in the BERT-based models. For the GPT2-based models on the Moth Radio Hour dataset, results are not consistent due to the weaker effect of brain removal.

**Effects across Linguistic Subfields.** We further investigate this effect by analyzing performance across different linguistic subfields: syntax, semantics, discourse, reasoning, and morphology. As shown in Figure 3B, the Brain Misaligned model consistently underperforms the Brain Preserving model in discourse, morphology, reasoning, semantics, and syntax tasks. This suggests that brain alignment is particularly important for supporting linguistic competence. Statistical tests on individual model and dataset combinations reveal significant differences in the majority of model-dataset combinations, even though the averaged results are not statistically significant. In particular, BERT trained on Harry Potter reveals statistical significance across all linguistic subfields. BERT trained on Moth Radio Hour and Llama trained on Harry Potter on the semantics and syntax subfields, while Llama trained on Moth Radio Hour on the syntax and morphology subfields.

**Effects across Linguistic Phenomena.** To gain finer-grained insights, we analyzed results based on specific linguistic phenomena, focusing on those represented by more than five datasets. As shown in Figure 4, we found that for the majority of tasks the Brain Preserving models are better than the Brain Misaligned models, providing further evidence that brain alignment is crucial for these phenomena. Examples for these linguistic phenomena can be found in Table 2.

## 4.3 Further Validation via Brain-Tuning

We conducted a complementary analysis by introducing a Brain Tuned model, in which, contrary to the Brain Misaligned model, the brain alignment capabilities were intentionally increased during training.

We then compared this new model with the Brain Preserving model. The analysis reveals that in every experimental setting, the Brain Tuned model consistently outperforms the Brain Preserving model with a statistically significant difference. This result suggests that increasing brain alignment translates into a general improvement in linguistic competence. Averaged results across model and dataset combinations are shown in Figure 5A. Figure 5B shows the averaged results across linguistic subfield, showing a statistically significant results for the syntax and semantic subfield, and Figure 6 shows the differences across linguistic phenomena, showing statistically significant results for two tasks. Detailed results for each of these combinations are available in the Appendix D, E, F,G,H, I.

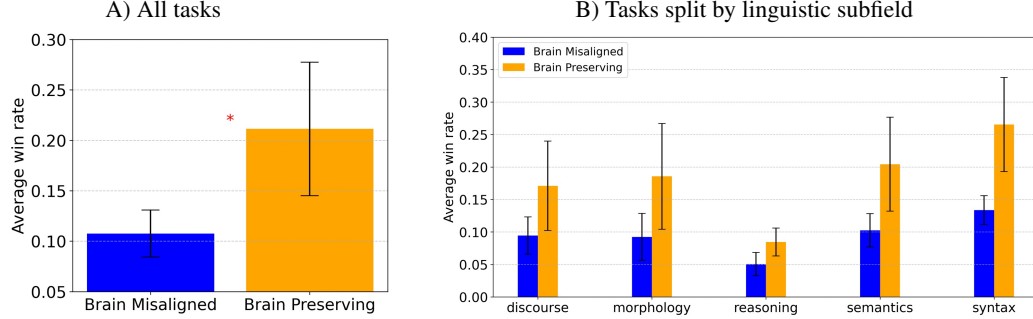

Figure 3: Average win rate and standard error across models and dataset combinations of the Brain Misaligned and Brain Preserving models across tasks (Left) and across different linguistic subfields (Right). The average win rate indicates how often each model outperforms its counterpart across model and dataset combinations. The Brain Preserving model significantly outperforms the Brain Misaligned model ($p < 0.05$, Wilcoxon signed-rank test) (Left). This result suggests that removing brain alignment impairs linguistic competence. The Brain Preserving model shows a higher win rate in all the linguistic subfield, in particular for semantics and syntax (Right), even if the differences are not statistically significant (assessed using Wilcoxon signed-rank test with Holm-Bonferroni correction), because of unique differences across model-dataset combinations.

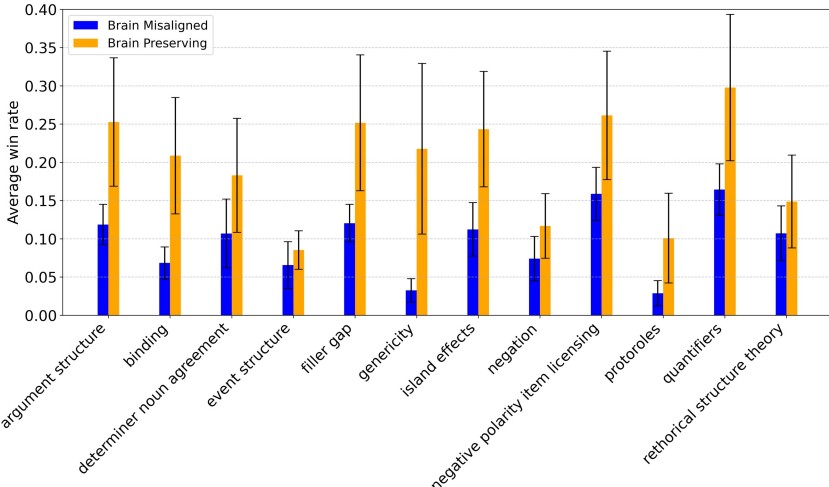

Figure 4: Average win rate with standard error across model and dataset combinations, across various linguistic phenomena for the Brain Misaligned and Brain Preserving models. Each bar represents the average win rate for a specific linguistic phenomenon, with error bars indicating standard error. Brain Preserving models tend to outperform Brain Misaligned models in the majority of tasks. Some concrete examples of the linguistic tasks are provided in the Table 2.

We also compared the Brain Tuned model directly with the original Pretrained model (i.e., the base model before any alignment intervention). While the Brain Tuned model showed an advantage in the majority of experimental settings, the Pretrained model maintained stronger performance in some settings. Results are reported in the Appendix D, E, F G H, I.

## 5 DISCUSSION

To investigate the importance of brain alignment in language models, we designed two models: the Brain Misaligned model, which is intended to remove brain alignment while preserving language modeling capabilities, and the Brain Preserving model, which accounts for potential confounders.

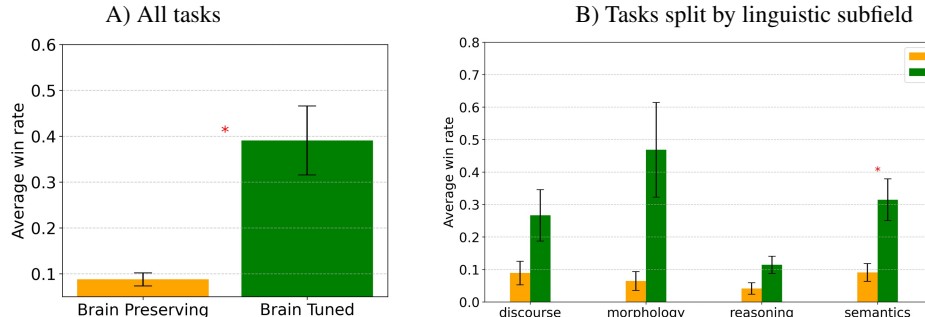

Figure 5: Average win rate and standard error across models and dataset combinations of the Brain Preserving and Brain Tuned models across tasks (Left) and across different linguistic subfields (Right). The Brain Tuned model significantly outperforms the Brain Preserving model ($p < 0.05$, Wilcoxon signed-rank test) (Left). This result suggests that improving the brain alignment lead to performance gains in linguistic competence. The Brain Tuned model shows a higher win rate in the discourse, morphology, reasoning, semantics and syntax subfield (Right) and significantly higher in semantics and syntax ($p < 0.05$, Wilcoxon signed-rank test with Holm-Bonferroni correction), suggesting that improving brain alignment affects semantics and syntax tasks.

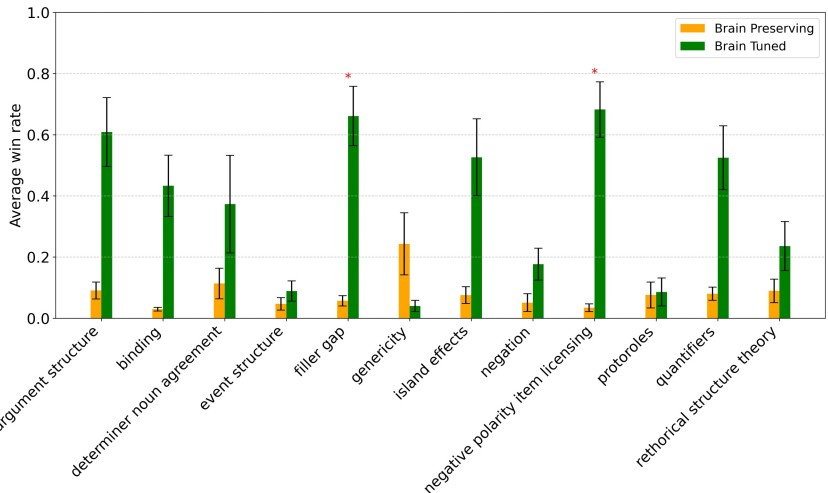

Figure 6: Average win rate with standard error across model and dataset combinations, across various linguistic phenomena for the Brain Tuned and Brain Preserving models. Each bar represents the average win rate for a specific linguistic phenomenon, with error bars indicating standard error. Brain Tuned models tend to outperform Brain Preserving models in the majority of tasks, with statistically significant difference ($p < 0.05$, Wilcoxon signed-rank test with Holm-Bonferroni correction) for `filler gap` and `negative polarity item licensing`. Some concrete examples of the linguistic tasks are provided in the Table 2.

We showed that the Brain Misaligned model has weak alignment with brain recordings, while the Brain Preserving model exhibits stronger alignment, particularly in language-related regions of interest. The contrast between the two models, across multiple experimental settings, reveals that the difference in alignment is especially pronounced in these areas.

We further evaluated the linguistic competence of these models to reveal the functional importance of brain alignment. Our results demonstrate that Brain Misaligned models perform worse than Brain Preserving models on linguistic tasks, across multiple model-dataset combinations, supporting the hypothesis that brain alignment is crucial for maintaining linguistic competence. Across multiple experimental settings the performance drop is particularly evident in tasks related to the semantic and syntactic subfield, although there are unique differences in every experimental setting.

We extended this investigation by introducing a Brain Tuned model, designed to increase brain alignment. The results of this intervention further strengthen our core argument. We found that the Brain Tuned model systematically outperformed the Brain Preserving model in all experimental settings and in particular in semantic and syntax tasks. In many model-dataset combinations the Brain Tuned model outperform the pretrained model on linguistic competence highlighting the relevance of brain-related signal for improving those competences.

These findings, across multiple model-dataset combinations, suggest that brain-aligned information plays a key role in supporting performance on linguistic tasks. It is important to note that the absence of statistically significant differences for other linguistic subfields or phenomena does not imply that brain alignment is unimportant for those tasks.

**Limitations.** Our study has three main limitations. Firstly, the benchmark used to assess linguistic competence, while extensive, is not exhaustive. There are many additional datasets available that could be included in future evaluations (Wang et al., 2018; 2019). Moreover, some linguistic subfields (e.g., discourse) and specific linguistic phenomena are represented by only a few datasets. As a result, the observed behavior of the Brain Misaligned and Brain Preserving models may be influenced by the limited coverage and distribution of tasks in certain categories. Secondly, our results are based on limited fMRI datasets. While these widely studied datasets offer extensive per-participant data, findings may still be specific to their characteristics. We designed our experiments using cross-validation, testing on held-out data and across multiple participants to improve generalizability. However, results might differ with different types of linguistic stimuli. Expanding to more datasets, languages, or cognitive tasks would be an important next step. Thirdly, while the "Brain Misaligned" model does show a clear overall worse performance, there are differences across linguistic subfields depending on the model-dataset combination. Datasets and models can contain different types of information related to linguistic subfields. Nevertheless, our results are informative in demonstrating the effectiveness of our methodology and in highlighting the importance of the emergent brain alignment ability of language models.

## 6 CONCLUSION

We designed a direct approach to investigate the necessity of brain alignment in pretrained language models. Specifically, we introduced two models: the Brain Misaligned model and the Brain Preserving model. When used together, they allow us to isolate and control for the effect of brain alignment on downstream linguistic competence.

We evaluated these models on more than 200 datasets spanning various linguistic subfields, including semantics, syntax, morphology, discourse, and reasoning, as well as a broad range of linguistic phenomena. Our results revealed a significant drop in linguistic competence, particularly on semantic and syntactic tasks, for the Brain Misaligned model, suggesting that brain alignment plays a critical role in downstream linguistic performance. This conclusion is further supported by our complementary finding that a Brain Tuned model, optimized to increase alignment, consistently outperformed the Brain Preserving model particularly in those tasks.

These findings are highly relevant to the natural language processing literature. Previous studies have explored why brain alignment emerges during pretraining, pointing to possible contributing factors and suggesting that if this alignment emerges, it may reflect shared information acquisition between artificial and biological neural networks.

Our work contributes a new dimension to this discussion: we not only ask why brain alignment emerges, but also whether it is important for linguistic competence. Our results provide initial evidence that brain alignment is functionally important, motivating future research in this area.

Moreover, our methodology provides a general framework for assessing the causal role of emergent properties, as brain alignment, in language models. Future work could apply our methodology to different models, exploring other datasets, or extending the approach to assess the necessity of alignment-related capabilities across different modalities (e.g., speech, image) or neural architectures.

ACKNOWLEDGMENTS

Work by Gabriele Merlin was supported by the CS@max planck graduate center.

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

## A  LINGUISTIC COMPETENCE BENCHMARK

We evaluated the linguistic competence of language models using classifier-based probing on more than 200 datasets collected from various sources and included in the Holmes benchmark (Waldis et al., 2024). The benchmark covers datasets spanning a wide range of linguistic phenomena and subfields, including syntax, morphology, semantics, reasoning, and discourse. A comprehensive list of linguistic phenomena and their corresponding subfields is provided in Table 3. For the evaluation, we use the `flash-holmes` version of the benchmark, which is designed to reduce computational cost while maintaining precision in assessing language model performance (see Waldis et al. (2024) for details). Examples of tasks associated with the linguistic phenomena and linguistic subfield used in our study are reported in Tables 1 and 2 and illustrated in Figure 4.

## B  BRAIN MAPPING HEAD

To predict the fMRI recordings corresponding to each TR, we use a linear function, regularized with a ridge penalty, that maps model representations to fMRI space, specifically targeting voxels with an estimated noise ceiling $> 0.05$ located in language-related regions of interest (Figure 2D). This function is trained in a cross-validated way and evaluated on held-out data. The ridge penalty

Table 1: Examples for linguistic subfields from Waldis et al. (2024). The relevant part of the example for the specific label is underlined.

| Type | Phenomena | Example | Label |
|---|---|---|---|
| **Morphology** | Subject-Verb Agreement | *And then, the cucumber was hurled into the air.* | Correct |
| | | *And then, the cucumber were hurled into the air.* | Wrong |
| **Syntax** | Part-of-Speech | *And then, the cucumber was hurled into the air.* | NN (Noun Singular) |
| **Semantics** | Semantic Roles | *And then, the cucumber was hurled into the air.* | Direction |
| **Reasoning** | Negation | *And then, the cucumber was hurled into the air.* | No Negation |
| **Discourse** | Node Type in Rhetorical Tree | *And then, the cucumber was hurled into the air.* | Satellite |

Table 2: Examples for selected linguistic phenomena from Waldis et al. (2024). The asterisk (*) indicates the correct option when applicable.

| Phenomena | Illustrative Example |
|---|---|
| *argument-structure* | Most cashiers are disliked*/flirted. |
| *binding* | Carlos said that Lori helped him*/himself. |
| *determiner noun agreement* | Craig explored that grocery store*/stores. |
| *event structure* | Give them to a library or burn them. ⇒ Distributive |
| *filler-gap* | Brett knew what many waiters find.*/Brett knew that many waiters find. |
| *genericity* | I assume you mean the crazy horse memorial. ⇒ Not Dynamic |
| *island-effects* | Which bikes is John fixing?*/Which is John fixing bikes? |
| *antonym negation* | It was not*/really hot, it was cold. |
| *negative polarity item licensing* | Only/Even Bill would ever complain. |
| *semantic proto-roles* | These look fine to me. ⇒ Exists as physical |
| *quantifiers* | There aren't many*/all lights darkening. |
| *rhetorical structure theory* | The statistics quoted by the " new " Census Bureau report ⇒ Elaboration |
| *subject-verb agreement* | A sketch of lights does not*/do not appear. |

is selected via nested cross-validation. For each participant, we train four functions, each using three of the four fMRI subsets for training and the remaining one for testing. To generate model representations, we average the token embeddings corresponding to each TR, and construct the input by concatenating the embeddings from the current TR with those from the previous five TRs. The features of the words presented in the previous TRs are included to account for the lag in the hemodynamic response that fMRI records. Because the response measured by fMRI is an indirect consequence of brain activity that peaks about 6 seconds after stimulus onset, predictive methods commonly include preceding time points (Nishimoto et al., 2011; Wehbe et al., 2014a; Huth et al., 2016). This allows for a data-driven estimation of the hemodynamic response functions (HRFs) for each voxel, which is preferable to assuming one because different voxels may exhibit different HRFs.

Table 3: List of linguistic phenomena and their corresponding subfields in the Holmes benchmark.

| linguistic phenomena | subfield | linguistic phenomena | subfield |
| --- | --- | --- | --- |
| next sentence prediction | discourse | semantic odd man out | semantics |
| rhetorical structure theory | discourse | word sense | semantics |
| sentence order | discourse | word content | semantics |
| discourse connective | discourse | coordination inversion | semantics |
| coreference resolution | discourse | object animacy | semantics |
| bridging | discourse | event structure | semantics |
| irregular forms | morphology | factuality | semantics |
| subject-verb agreement | morphology | complex words | semantics |
| determiner noun agreement | morphology | genericity | semantics |
| anaphor agreement | morphology | metaphor | semantics |
| age comparison | reasoning | named entity labeling | semantics |
| negation | reasoning | negative polarity item licensing | semantics |
| speculation | reasoning | argument structure | syntax |
| multi-hop composition | reasoning | bigram-shift | syntax |
| property conjunction | reasoning | binding | syntax |
| object comparison | reasoning | tree-depth | syntax |
| antonym negation | reasoning | case | syntax |
| encyclopedic composition | reasoning | subject-verb agreement | syntax |
| taxonomy conjunction | reasoning | anaphor agreement | syntax |
| always never | reasoning | top-constituent-task | syntax |
| object gender | semantics | subject number | syntax |
| passive | semantics | deoncausative-inchoative alternation | syntax |
| protoroles | semantics | control / raising | syntax |
| quantifiers | semantics | ellipsis | syntax |
| synonym-/antonym-detection | semantics | sentence length | syntax |
| verb dynamic | semantics | filler gap | syntax |
| semantic role labeling | semantics | readability | syntax |
| sentiment analysis | semantics | island effects | syntax |
| time | semantics | local attractor | syntax |
| subject animacy | semantics | part-of-speech | syntax |
| subject gender | semantics | object number | syntax |
| tense | semantics | constituent parsing | syntax |
| relation classification | semantics | negative polarity item licensing | syntax |

## C  NOISE CEILING ESTIMATION

To assess the signal quality of the fMRI data, we estimated noise ceiling values, which quantify the proportion of variance that could be explained by an ideal data-generating model. This method involves predicting the fMRI activity of a target participant using linear models trained on data from another participant. For a more detailed explanation, refer to Schrimpf et al. (2021). Estimating the noise ceiling is particularly useful given the inherently high level of noise in fMRI data.

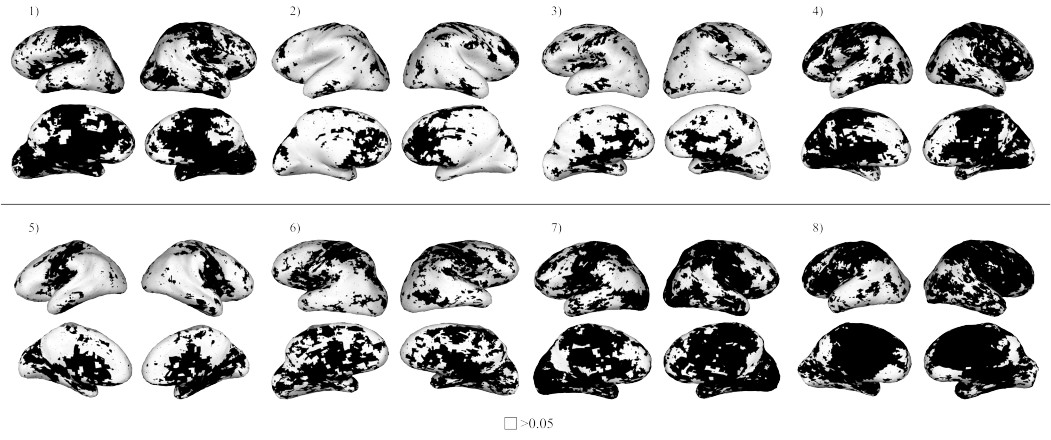

Figure 7: Voxel-wise estimated noise ceiling values for participants included in Harry Potter dataset (Wehbe et al., 2014a). To exclude noisy voxels, we selected, for each participant, those with noise ceiling estimates above 0.05.

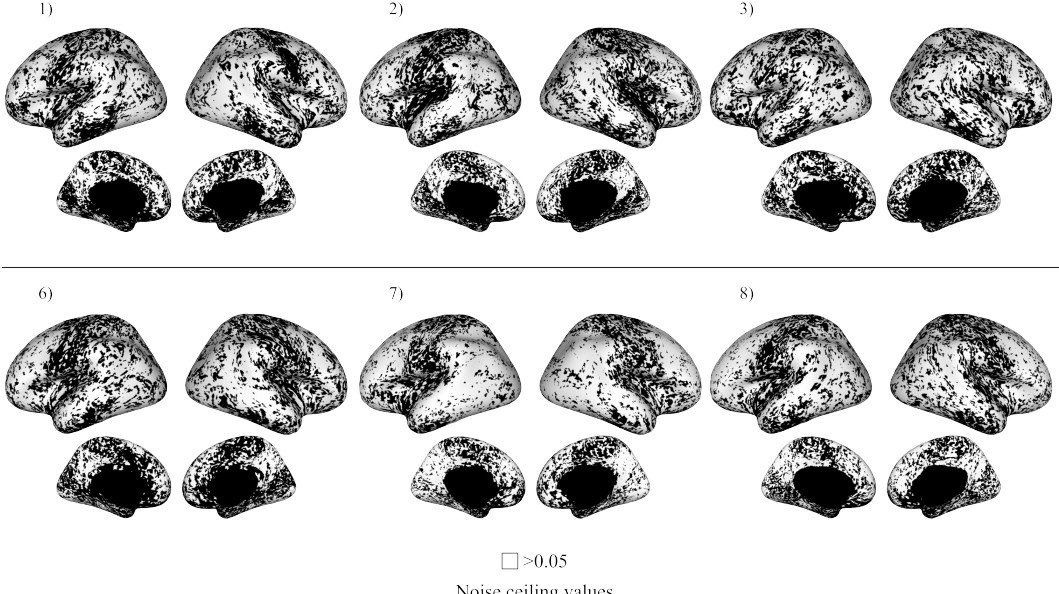

Figure 8: Voxel-wise estimated noise ceiling values for participants included in Moth Radio Hour dataset (Deniz et al., 2019). To exclude noisy voxels, we selected, for each participant, those with noise ceiling estimates above 0.05.

## D   BERT MISALIGNMENT ON HARRY POTTER DATASET

We report the brain alignment results for Brain Misaligned and Brain Preserving trained with data from each participant in Figure 9, as well as a quantitative summary in Figure 10. Figure 11 report the quantitative summary for brain alignment for the Brain Tuned model compared to Brain Preserving model. Results for the Holmes benchmark for all the comparisons are reported in Figure 12, 13, 14, 15, 16, 17.

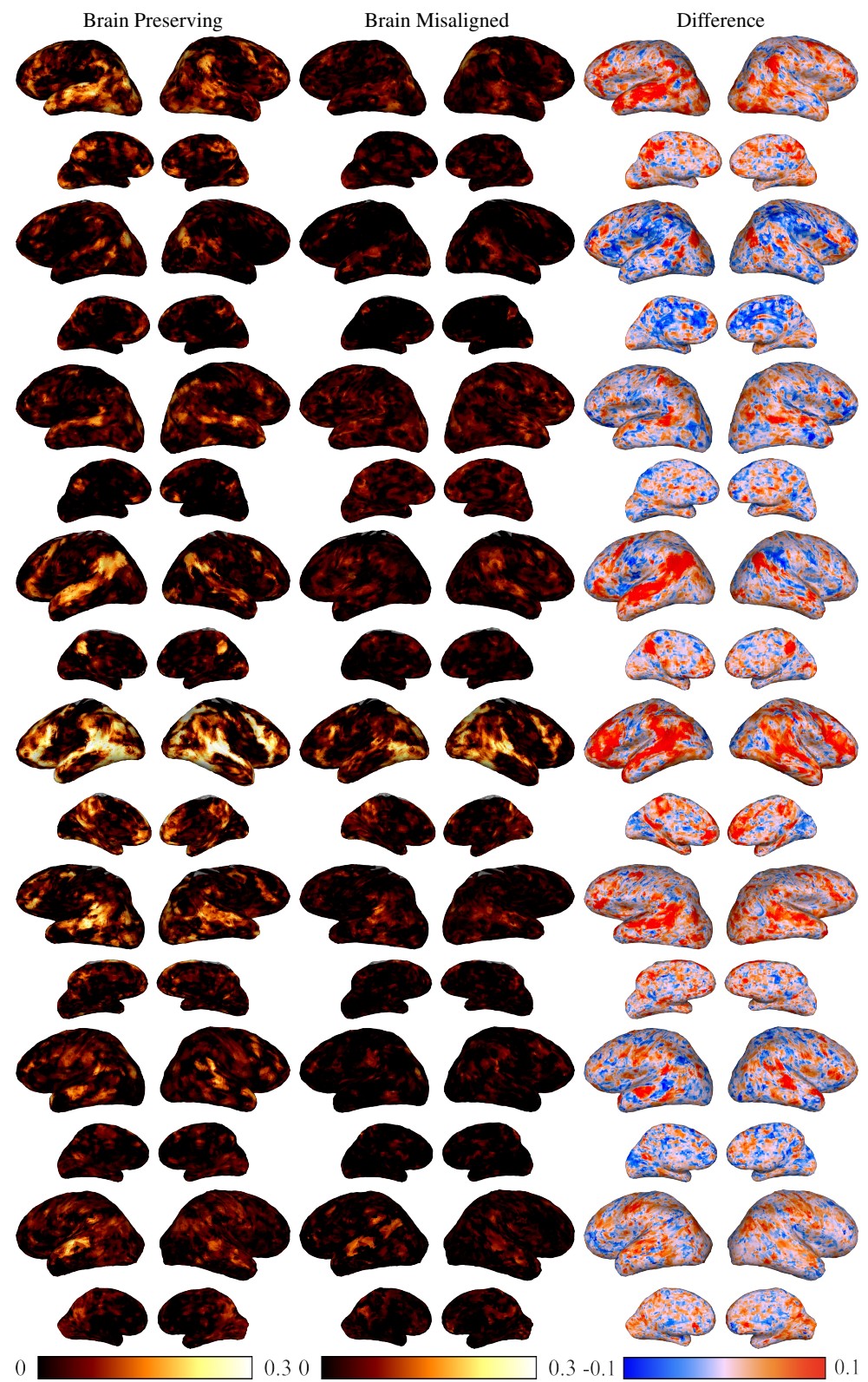

Figure 9: Performances of BERT-based Brain Misaligned and Brain Preserving models on the Harry Potter dataset at the brain alignment task. Brain plots show voxel-wise Pearson correlations between model activations and brain responses for each subject. The left column displays results for the Brain Preserving model, the center column for the Brain Misaligned model, and the right column shows their difference (Preserving minus Misaligned). Warmer colors indicate stronger alignment with brain activity. These results illustrate the distribution of brain alignment across subjects and highlight areas where brain misalignment has effects.

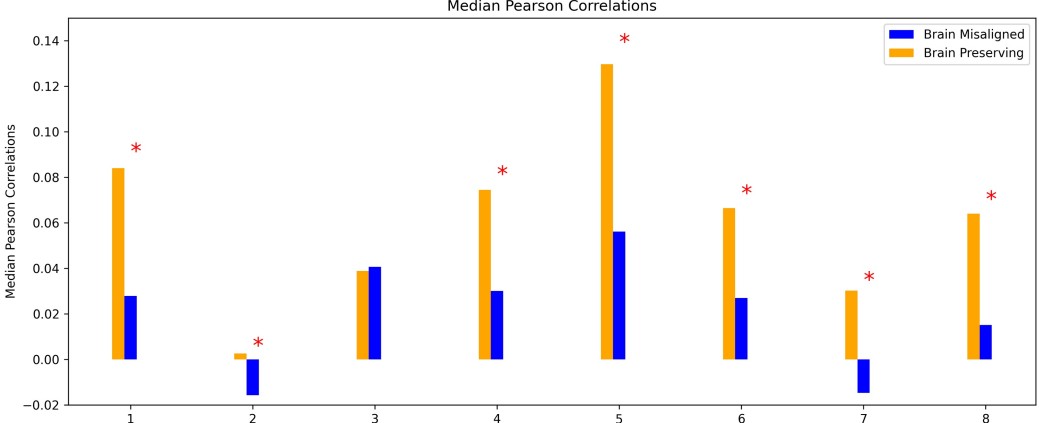

Figure 10: Median Pearson correlation for BERT-based models on the Harry Potter dataset for each participant. Brain Misaligned models perform significantly worse than Brain Preserving models for seven subjects ($p < 0.05$, indicated by * and assessed using the Wilcoxon signed-rank test).

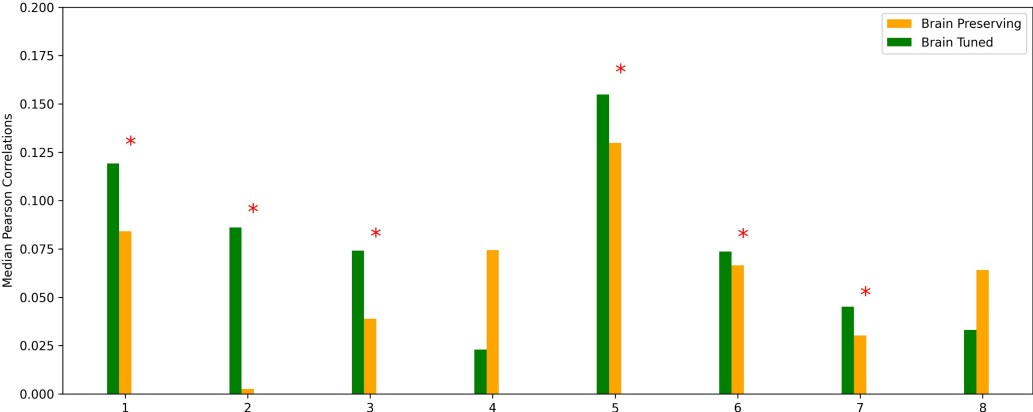

Figure 11: Median Pearson correlation for BERT-based models on the Harry Potter dataset for each participant. Brain Preserving models perform significantly worse than Brain Tuned models for six subjects ($p < 0.05$, indicated by * and assessed using the Wilcoxon signed-rank test).

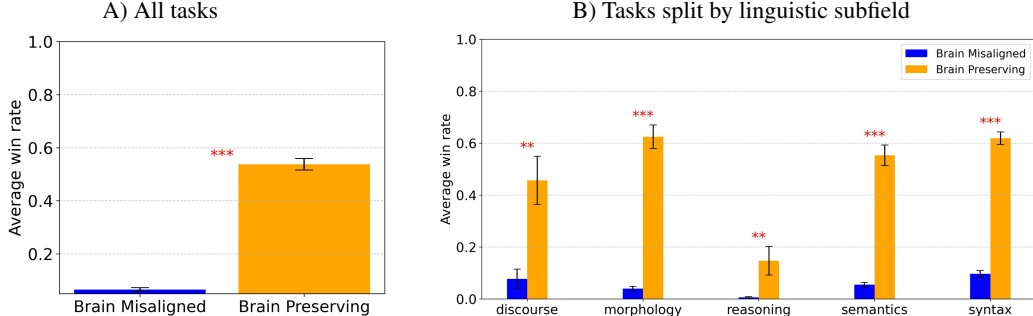

Figure 12: Average win rate and standard error of the BERT-based Brain Misaligned and Brain Preserving models on the Harry Potter dataset across participants and tasks (Left) and across different linguistic subfields (Right). The win rate indicates how often each model outperforms its counterpart across tasks and participants. The Brain Preserving model significantly outperforms the Brain Misaligned model ($p < 0.001$, indicated by ***), as assessed using a Wilcoxon signed-rank test (Left). This result suggests that removing brain alignment negatively influences linguistic competence. The Brain Preserving model shows a significantly higher win rate in all the linguistic subfield (Right) ($p < 0.05$, Wilcoxon signed-rank test with Holm-Bonferroni correction), suggesting that improving brain alignment affects all linguistic subfields.

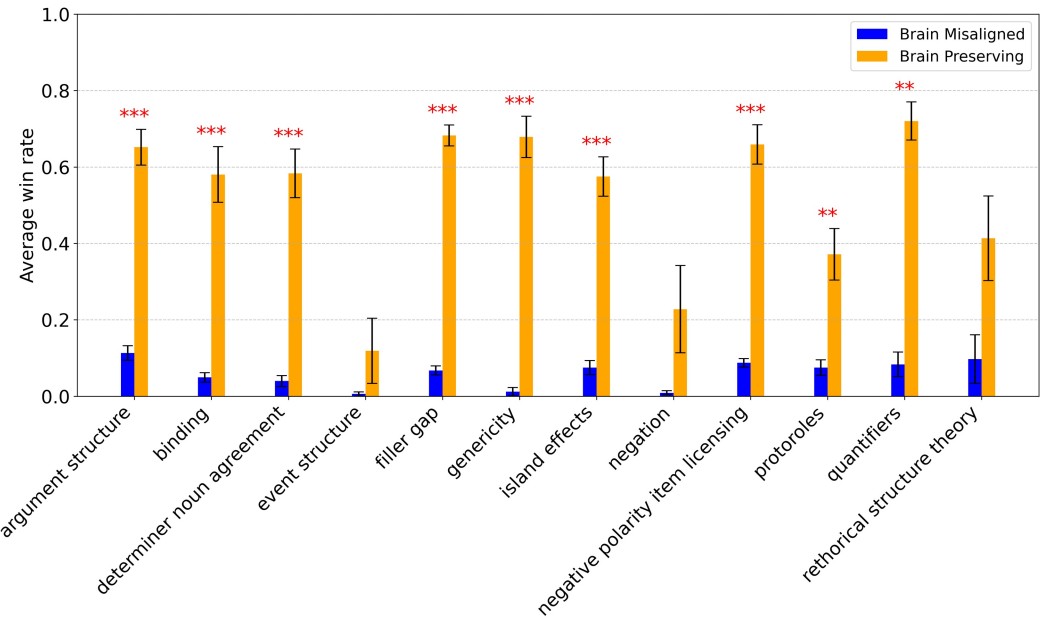

Figure 13: Average win rate with standard error across various linguistic phenomena for the BERT-based Brain Misaligned and Brain Preserving models on the Harry Potter dataset. Each bar represents the average win rate for a specific linguistic phenomenon, with error bars indicating standard error. Some concrete examples of the linguistic tasks are provided in the Table 2.

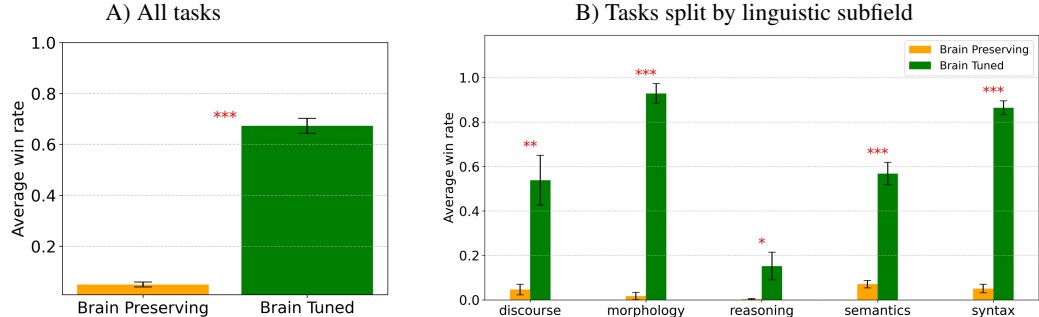

Figure 14: Average win rate and standard error of the BERT-based Brain Preserving and Brain Tuned models on the Harry Potter dataset across participants and tasks (Left) and across different linguistic subfields (Right). The win rate indicates how often each model outperforms its counterpart across tasks and participants. The Brain Tuned model significantly outperforms the Brain Preserving model ($p < 0.001$, indicated by ***), as assessed using a Wilcoxon signed-rank test (Left). This result suggests that improving brain alignment positively influences linguistic competence. The Brain Tuned model shows a higher win rate in the syntax, semantics, reasoning, morphology and discourse subfield (Right) and significantly higher for all linguistic subfields ($p < 0.05$, Wilcoxon signed-rank test with Holm-Bonferroni correction), suggesting that improving brain alignment affects all linguistic subfields.

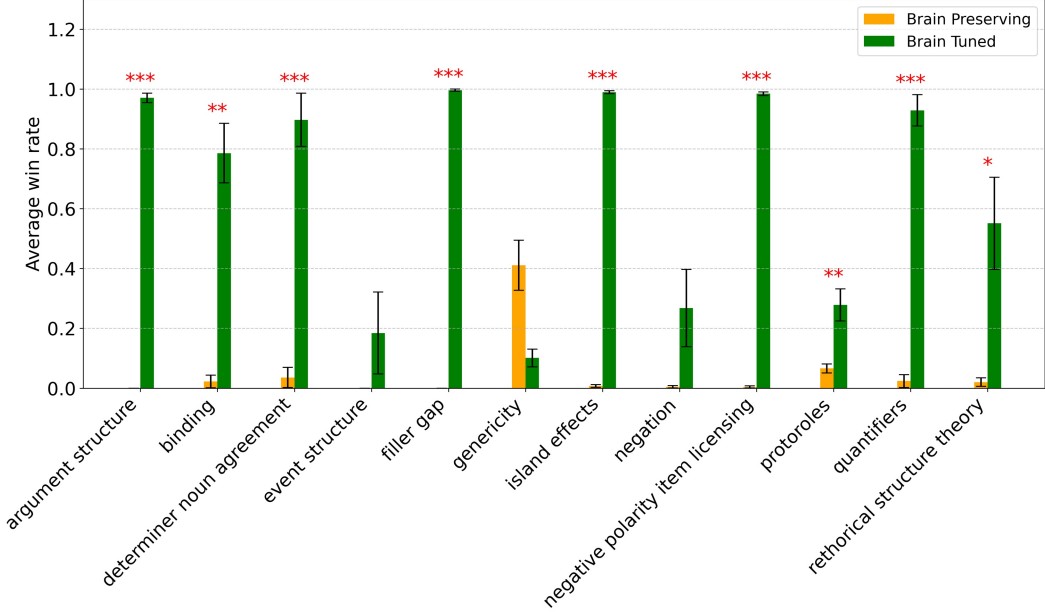

Figure 15: Average win rate with standard error across various linguistic phenomena for the BERT-based Brain Preserving and Brain Tuned models on the Harry Potter dataset. Each bar represents the average win rate for a specific linguistic phenomenon, with error bars indicating standard error. Some concrete examples of the linguistic tasks are provided in the Table 2.

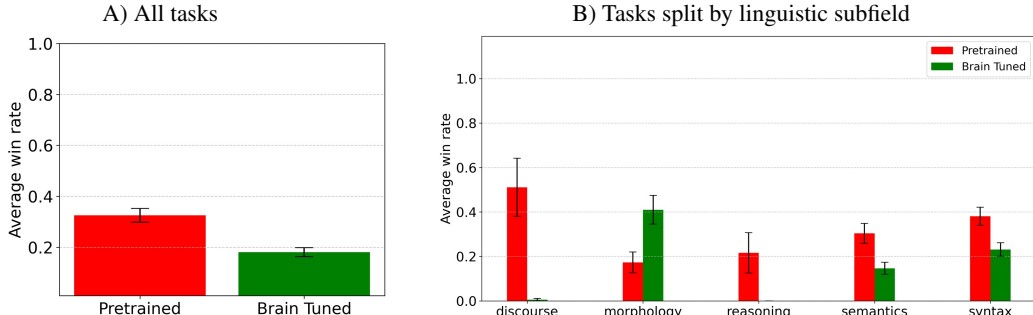

Figure 16: Average win rate and standard error of the BERT-based Brain Tuned and Pretrained models on the Harry Potter dataset across participants and tasks (Left) and across different linguistic subfields (Right). The win rate indicates how often each model outperforms its counterpart across tasks and participants. The Brain Tuned model shows a higher win rate in the morphology subfield (Right) (although Wilcoxon signed-rank test with Holm-Bonferroni correction reveal no significance), suggesting that improving brain alignment affects this linguistic subfield.

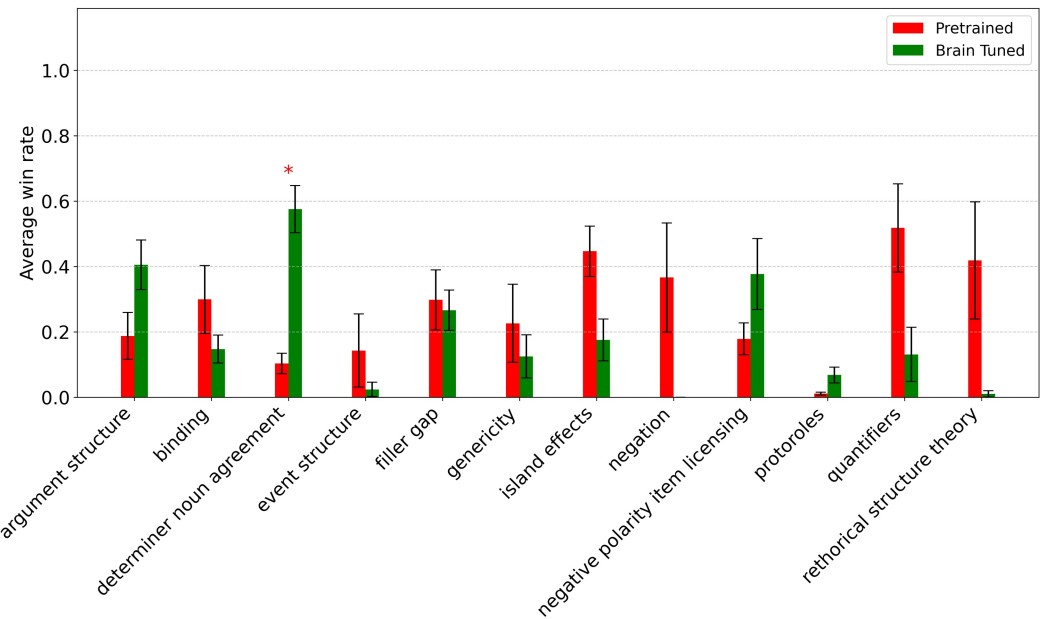

Figure 17: Average win rate with standard error across various linguistic phenomena for the BERT-based Brain Tuned and Pretrained models on the Harry Potter dataset. Each bar represents the average win rate for a specific linguistic phenomenon, with error bars indicating standard error. Some concrete examples of the linguistic tasks are provided in the Table 2.

## E    BERT MISALIGNMENT ON MOTH RADIO HOUR DATASET

We report the brain alignment results for Brain Misaligned and Brain Preserving trained with data from each participant in Figure 18, as well as a quantitative summary in Figure 19. Figure 20 report the quantitative summary for brain alignment for the Brain Tuned model compared to Brain Preserving model. Results for the Holmes benchmark for all the comparisons are reported in Figure 21, 22, 23, 24, 25, 26.

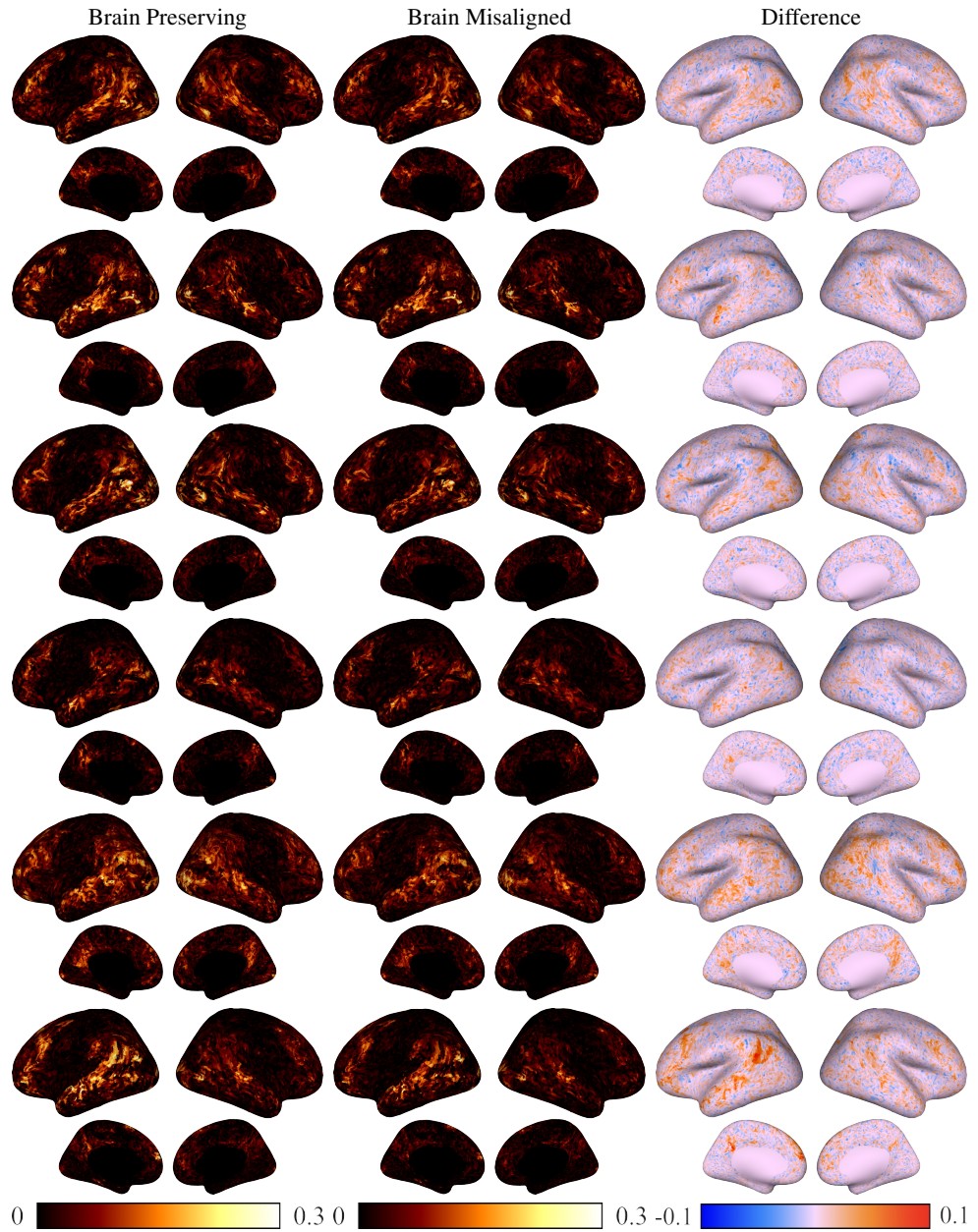

Figure 18: Performances of BERT-based Brain Misaligned and Brain Preserving models on the Moth Radio Hour dataset at the brain alignment task. Brain plots show voxel-wise Pearson correlations between model activations and brain responses for each subject. The left column displays results for the Brain Preserving model, the center column for the Brain Misaligned model, and the right column shows their difference (Preserving minus Misaligned). Warmer colors indicate stronger alignment with brain activity. These results illustrate the distribution of brain alignment across subjects and highlight areas where brain misalignment has effects.

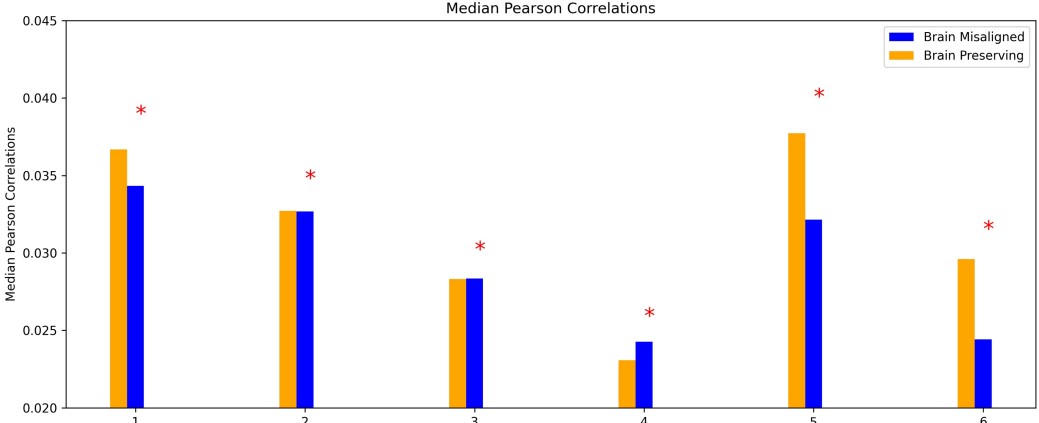

Figure 19: Median Pearson correlation for BERT-based models on the Moth Radio Hour dataset for each participant. Brain Misaligned models perform significantly worse than Brain Preserving models for six subjects ($p < 0.05$, indicated by * and assessed using the Wilcoxon signed-rank test).

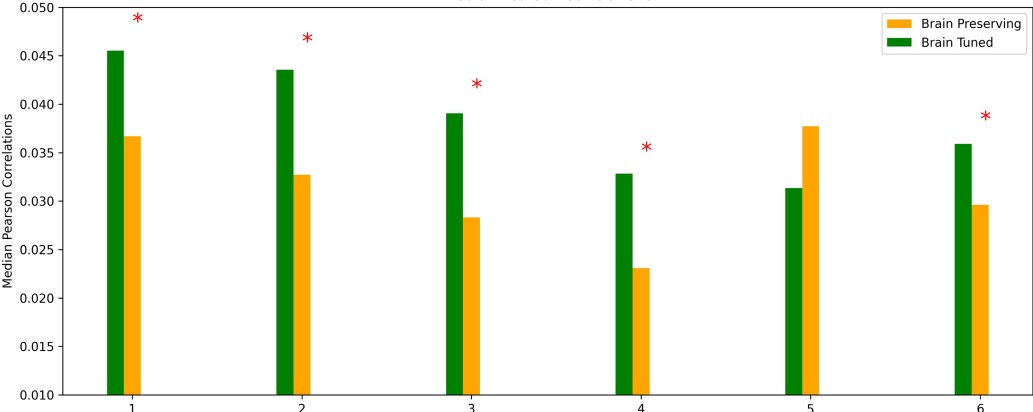

Figure 20: Median Pearson correlation for BERT-based models on the Moth Radio Hour dataset for each participant. Brain Preserving models perform significantly worse than Brain Tuned models for five subjects ($p < 0.05$, indicated by * and assessed using the Wilcoxon signed-rank test).

A) All tasks

B) Tasks split by linguistic subfield

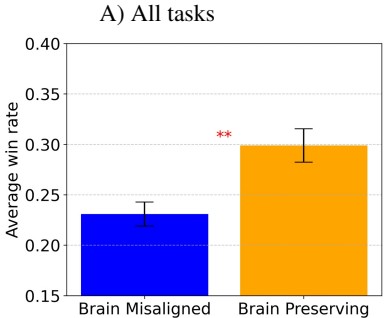
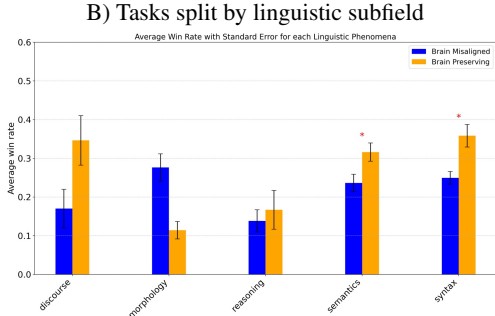

Figure 21: Average win rate and standard error of the BERT-based Brain Misaligned and Brain Preserving models on the Moth Radio Hour dataset across participants and tasks (Left) and across different linguistic subfields (Right). The win rate indicates how often each model outperforms its counterpart across tasks and participants. The Brain Preserving model significantly outperforms the Brain Misaligned model ($p < 0.01$, indicated by **), as assessed using a Wilcoxon signed-rank test (Left). This result suggests that removing brain alignment negatively influences linguistic competence. The Brain Preserving model shows a higher win rate in the syntax, semantics, reasoning and discourse subfield (Right) and significantly higher for syntax and semantics subfields ($p < 0.05$, Wilcoxon signed-rank test with Holm-Bonferroni correction), suggesting that removing brain alignment particularly affects syntax and semantic tasks.

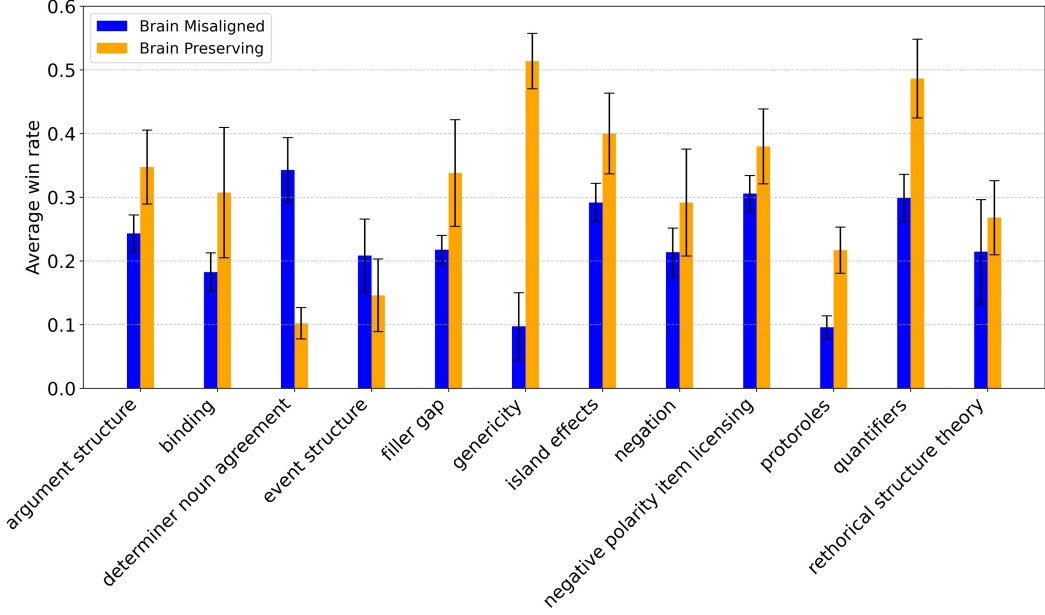

Figure 22: Average win rate with standard error across various linguistic phenomena for the BERT-based Brain Misaligned and Brain Preserving models on the Moth Radio Hour dataset. Each bar represents the average win rate for a specific linguistic phenomenon, with error bars indicating standard error. Brain Preserving models tend to outperform Brain Misaligned models, particularly in categories such as genericity and quantifiers. Some concrete examples of the linguistic tasks are provided in the Table 2.

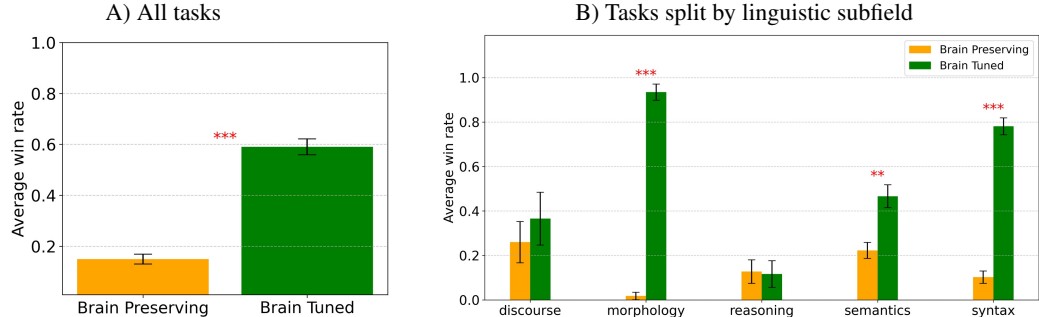

Figure 23: Average win rate and standard error of the BERT-based Brain Preserving and Brain Tuned models on the Moth Radio Hour dataset across participants and tasks (Left) and across different linguistic subfields (Right). The win rate indicates how often each model outperforms its counterpart across tasks and participants. The Brain Tuned model significantly outperforms the Brain Preserving model ($p < 0.001$, indicated by ***), as assessed using a Wilcoxon signed-rank test (Left). This result suggests that improving brain alignment positively influences linguistic competence. The Brain Tuned model shows a higher win rate in the syntax, semantics, morphology and discourse subfield (Right) and significantly higher for syntax, semantics and morphology subfields ($p < 0.05$, Wilcoxon signed-rank test with Holm-Bonferroni correction), suggesting that improving brain alignment affects those linguistic subfields.

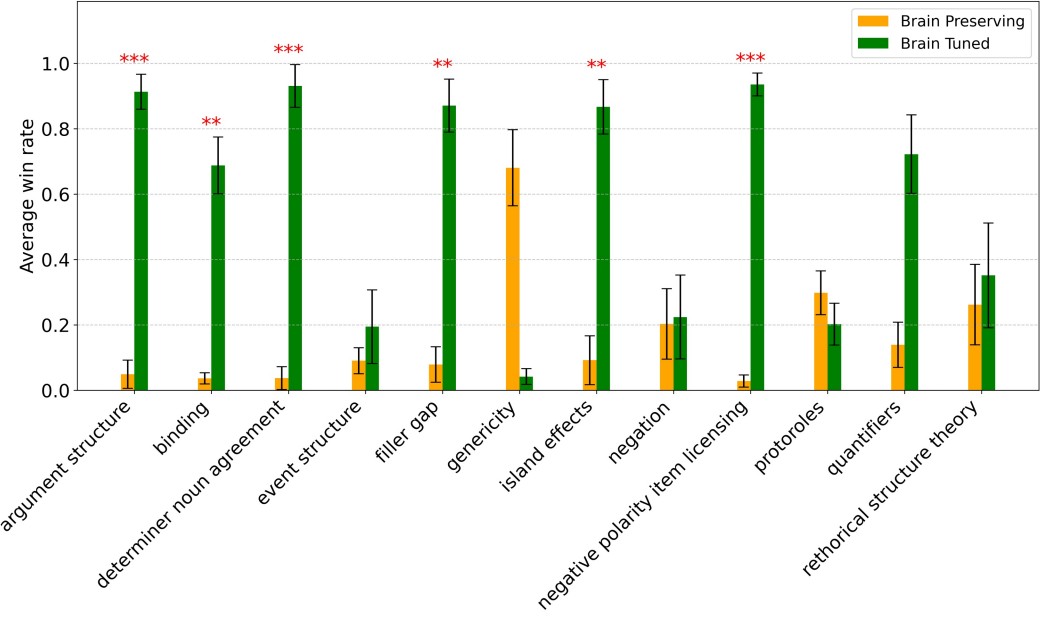

Figure 24: Average win rate with standard error across various linguistic phenomena for the BERT-based Brain Preserving and Brain Tuned models on the Moth Radio Hour dataset. Each bar represents the average win rate for a specific linguistic phenomenon, with error bars indicating standard error. Some concrete examples of the linguistic tasks are provided in the Table 2.

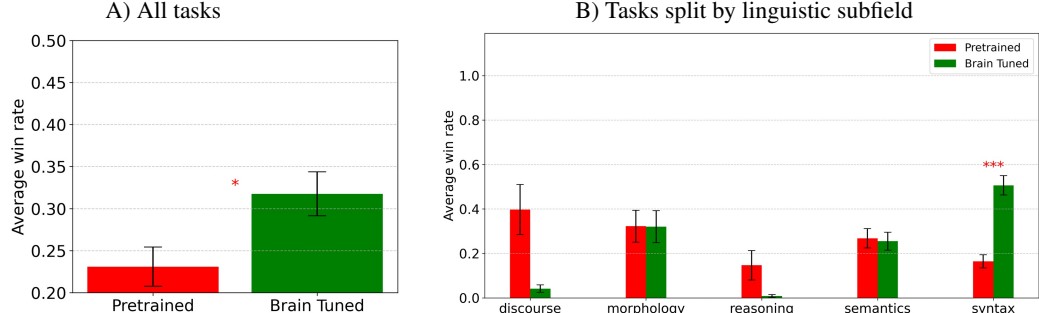

Figure 25: Average win rate and standard error of the BERT-based Brain Tuned and Pretrained models on the Moth Radio Hour dataset across participants and tasks (Left) and across different linguistic subfields (Right). The win rate indicates how often each model outperforms its counterpart across tasks and participants. The Brain Tuned model significantly outperforms the Pretrained model ($p < 0.05$, indicated by *), as assessed using a Wilcoxon signed-rank test (Left). This result suggests that improving brain alignment positively influences linguistic competence. The Brain Tuned model shows a higher win rate in the syntax subfield (Right) and significantly higher for syntax subfield ($p < 0.05$, Wilcoxon signed-rank test with Holm-Bonferroni correction), suggesting that improving brain alignment affects that linguistic subfield.

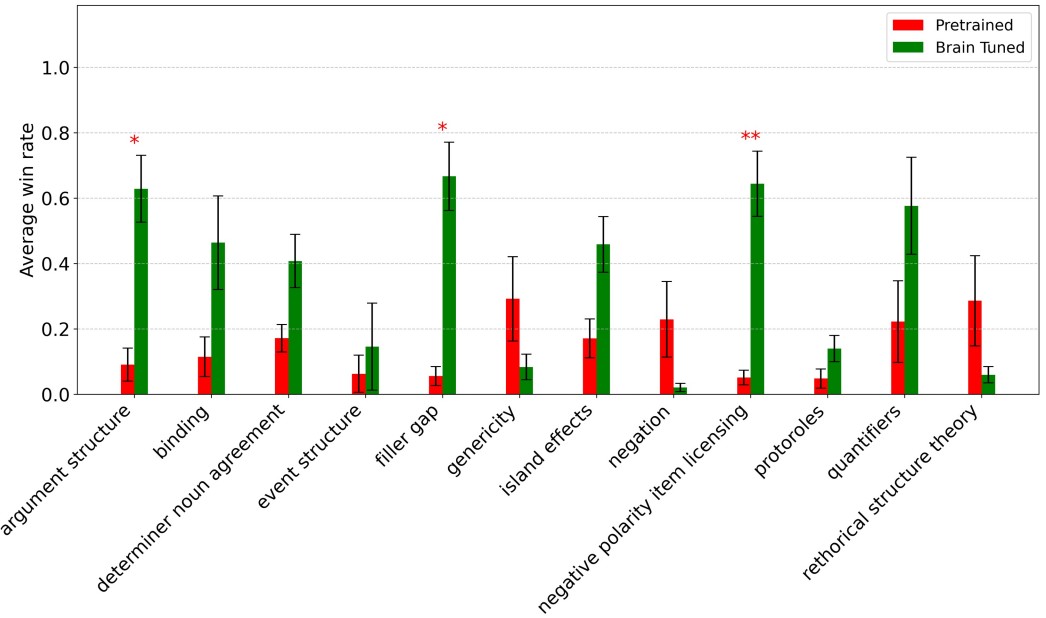

Figure 26: Average win rate with standard error across various linguistic phenomena for the BERT-based Brain Tuned and Pretrained models on the Moth Radio Hour dataset. Each bar represents the average win rate for a specific linguistic phenomenon, with error bars indicating standard error. Some concrete examples of the linguistic tasks are provided in the Table 2.

## F   GPT2 MISALIGNMENT ON HARRY POTTER BRAIN DATA RESULTS

We report the brain alignment results for Brain Misaligned and Brain Preserving trained with data from each participant in Figure 27, as well as a quantitative summary in Figure 28. Figure 29 report the quantitative summary for brain alignment for the Brain Tuned model compared to Brain Preserving model. Results for the Holmes benchmark for all the comparisons are reported in Figure 30, 31, 32, 33, 34, 35.

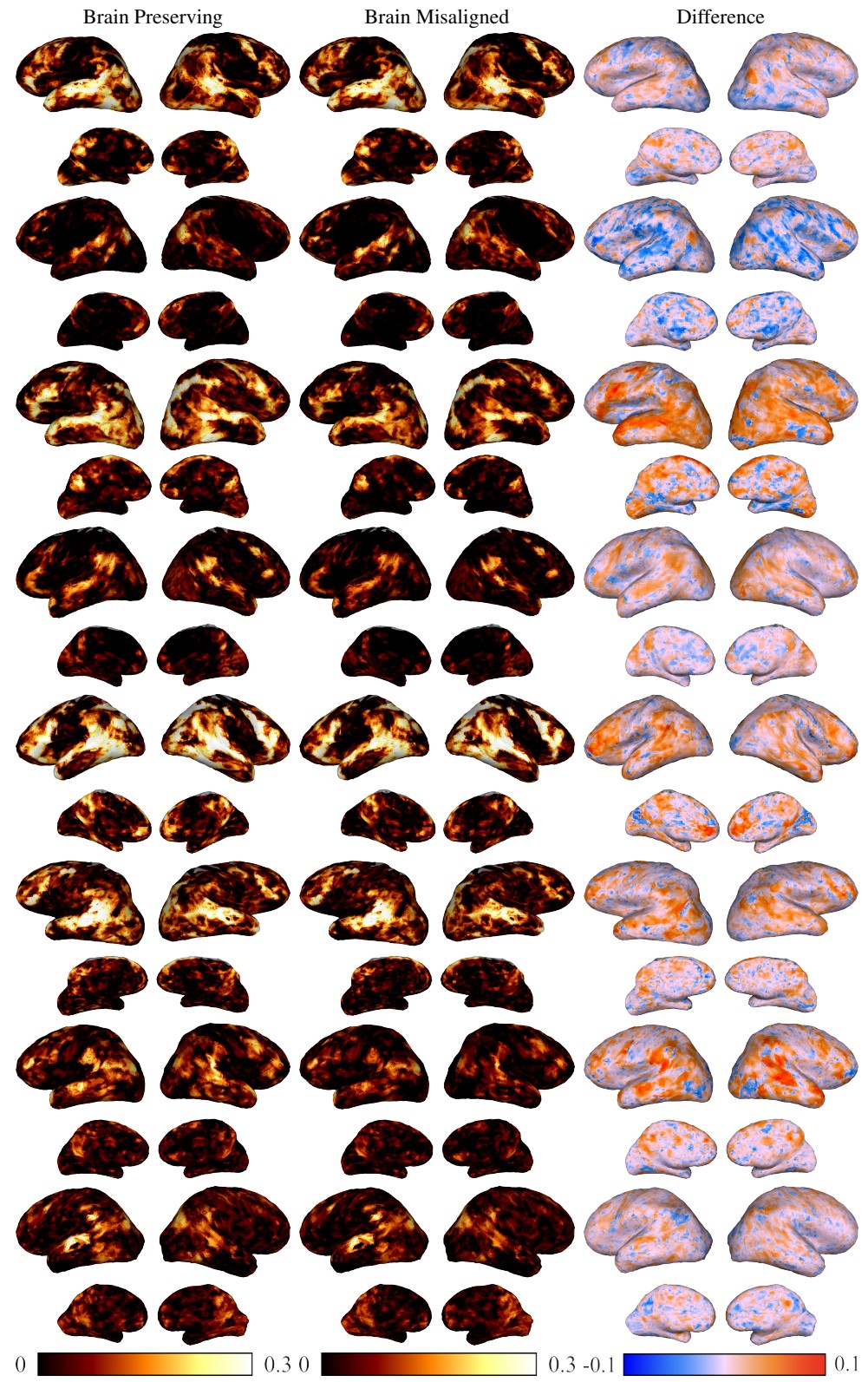

Figure 27: Performances of GPT2-based Brain Misaligned and Brain Preserving models on the Harry Potter dataset at the brain alignment task. Brain plots show voxel-wise Pearson correlations between model activations and brain responses for each subject. The left column displays results for the Brain Preserving model, the center column for the Brain Misaligned model, and the right column shows their difference (Preserving minus Misaligned). Warmer colors indicate stronger alignment with brain activity. These results illustrate the distribution of brain alignment across subjects and highlight areas where brain misalignment has effects.

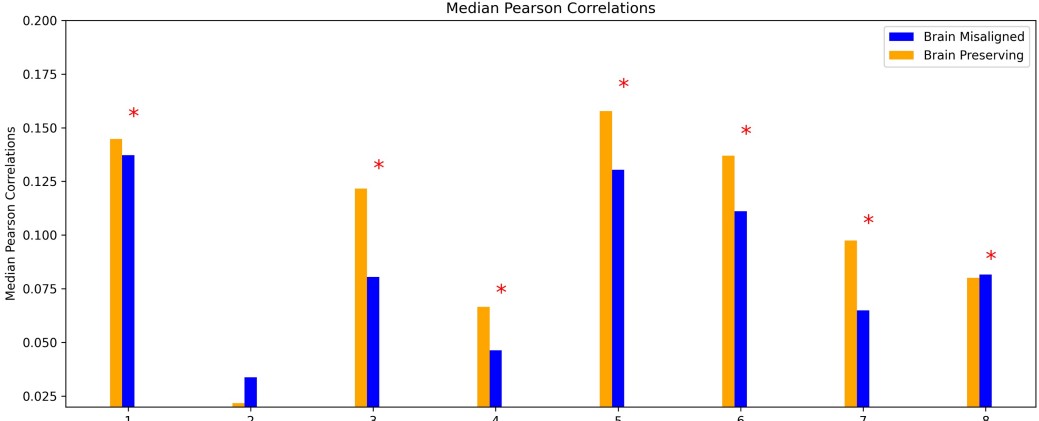

Figure 28: Median Pearson correlation for GPT2-based models on the Harry Potter dataset for each participant. Brain Misaligned models perform significantly worse than Brain Preserving models for seven subjects ($p < 0.05$, indicated by * and assessed using the Wilcoxon signed-rank test).

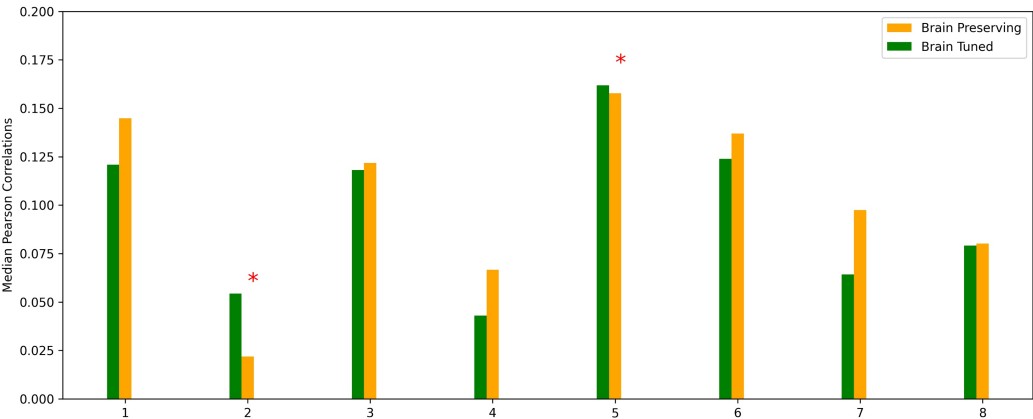

Figure 29: Median Pearson correlation for GPT2-based models on the Harry Potter dataset for each participant. Brain Preserving models perform significantly worse than Brain Tuned models for two subjects ($p < 0.05$, indicated by * and assessed using the Wilcoxon signed-rank test).

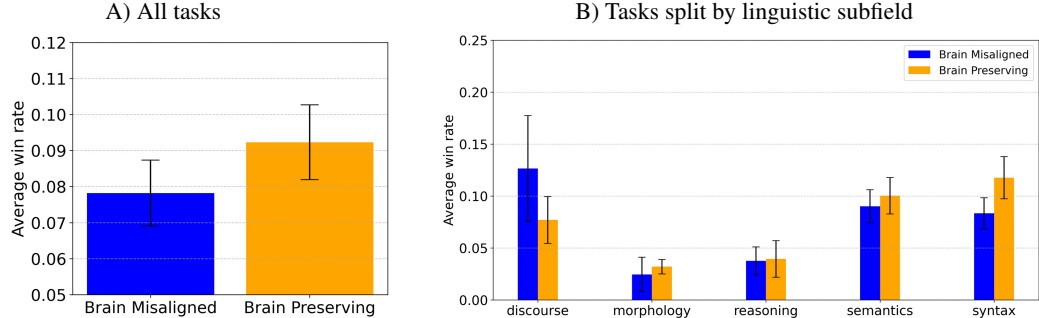

Figure 30: Average win rate and standard error of the GPT2-based Brain Misaligned and Brain Preserving models on the Harry Potter dataset across participants and tasks (Left) and across different linguistic subfields (Right). The win rate indicates how often each model outperforms its counterpart across tasks and participants. The Brain Preserving model outperforms the Brain Misaligned model (significance assessed using a Wilcoxon signed-rank test reveal $p = 0.055$) (Left). This result suggests that removing brain alignment negatively influences linguistic competence. The Brain Preserving model shows a higher win rate in particular in the semantics and syntax subfield (Right) (although Wilcoxon signed-rank test with Holm-Bonferroni correction reveal no significance), suggesting that removing brain alignment particularly affects semantics and syntax processing tasks.

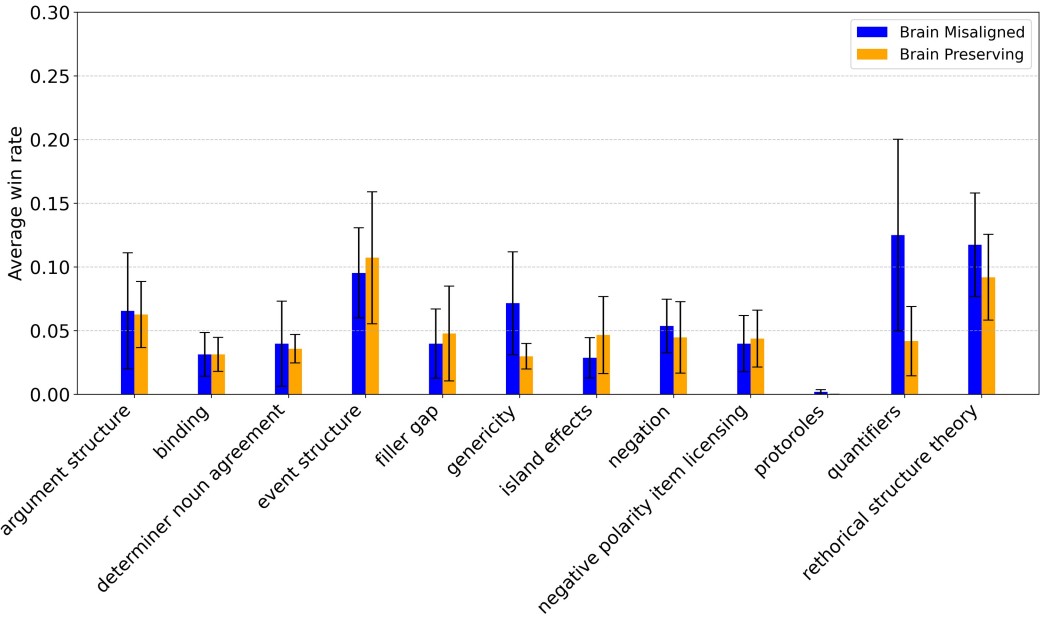

Figure 31: Average win rate with standard error across various linguistic phenomena for the GPT2-based Brain Misaligned and Brain Preserving models on the Harry Potter dataset. Each bar represents the average win rate for a specific linguistic phenomenon, with error bars indicating standard error. Some concrete examples of the linguistic tasks are provided in the Table 2.

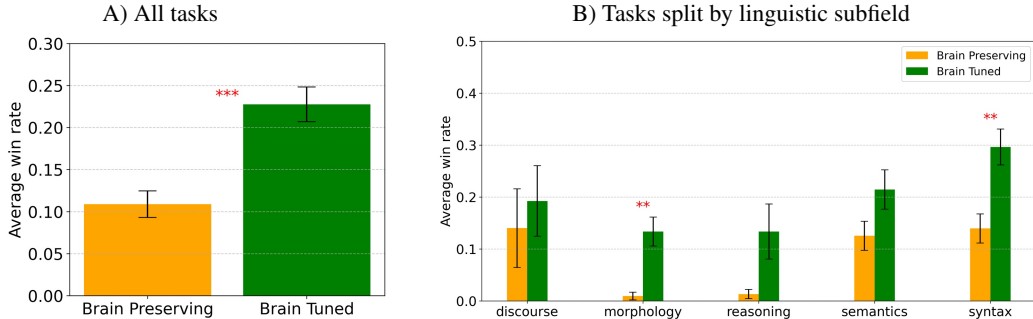

Figure 32: Average win rate and standard error of the GPT2-based Brain Preserving and Brain Tuned models on the Harry Potter dataset across participants and tasks (Left) and across different linguistic subfields (Right). The win rate indicates how often each model outperforms its counterpart across tasks and participants. The Brain Tuned model significantly outperforms the Brain Preserving model ($p < 0.001$, indicated by ***), as assessed using a Wilcoxon signed-rank test (Left). This result suggests that improving brain alignment positively influences linguistic competence. The Brain Tuned model shows a higher win rate in the syntax, semantics, reasoning, morphology and discourse subfield (Right) and significantly higher for syntax and morphology subfields ($p < 0.05$, Wilcoxon signed-rank test with Holm-Bonferroni correction), suggesting that improving brain alignment affects those linguistic subfields.

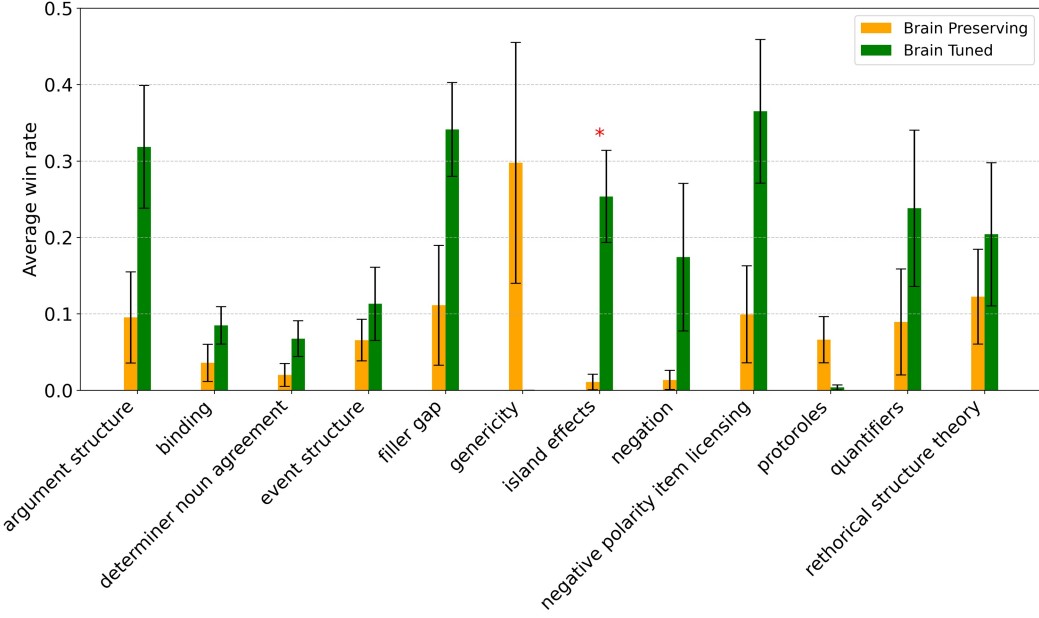

Figure 33: Average win rate with standard error across various linguistic phenomena for the GPT2-based Brain Preserving and Brain Tuned models on the Harry Potter dataset. Each bar represents the average win rate for a specific linguistic phenomenon, with error bars indicating standard error. Some concrete examples of the linguistic tasks are provided in the Table 2.

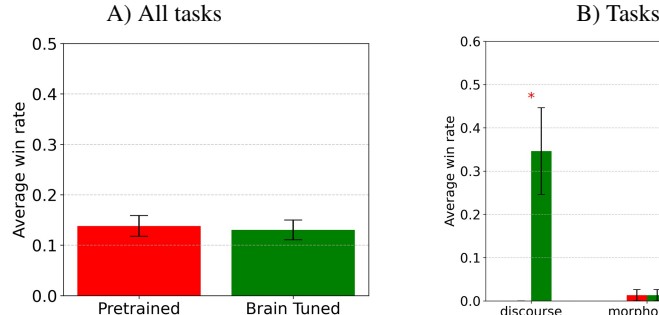

Figure 34: Average win rate and standard error of the GPT2-based Brain Tuned and Pretrained models on the Harry Potter dataset across participants and tasks (Left) and across different linguistic subfields (Right). The win rate indicates how often each model outperforms its counterpart across tasks and participants. The Brain Tuned model shows a higher win rate in the reasoning and discourse subfield (Right) and significantly higher for discourse subfield ($p < 0.05$, Wilcoxon signed-rank test with Holm-Bonferroni correction), suggesting that improving brain alignment affects that linguistic subfield.

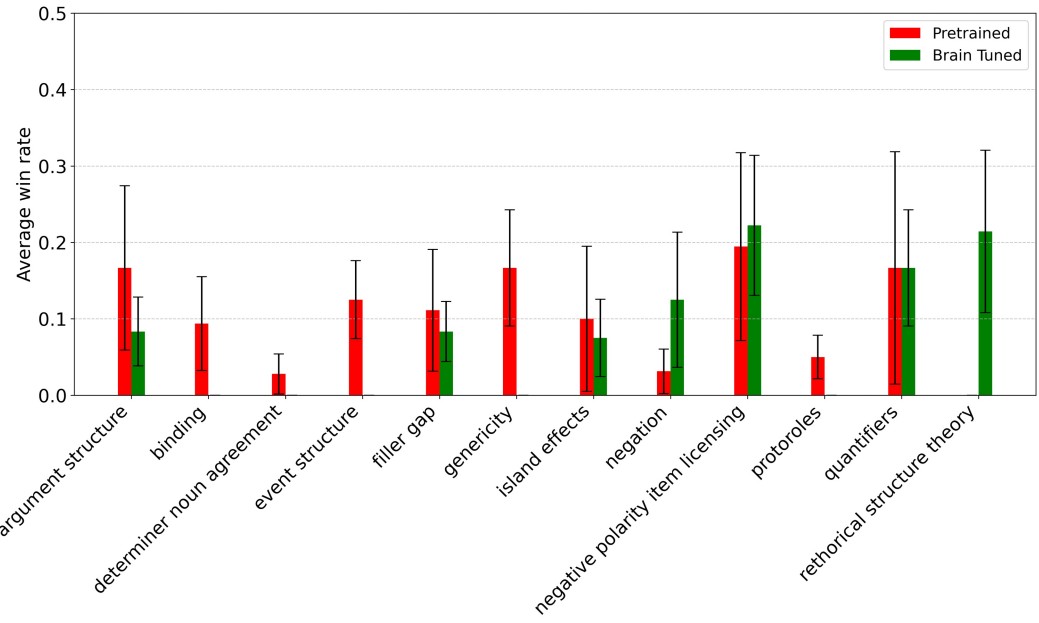

Figure 35: Average win rate with standard error across various linguistic phenomena for the GPT2-based Brain Tuned and Pretrained models on the Harry Potter dataset. Each bar represents the average win rate for a specific linguistic phenomenon, with error bars indicating standard error. Some concrete examples of the linguistic tasks are provided in the Table 2.

## G   GPT2 MISALIGNMENT ON MOTH RADIO HOUR DATASET

We report the brain alignment results for Brain Misaligned and Brain Preserving trained with data from each participant in Figure 36, as well as a quantitative summary in Figure 37. Figure 38 report the quantitative summary for brain alignment for the Brain Tuned model compared to Brain Preserving model. Results for the Holmes benchmark for all the comparisons are reported in Figure 39, 40, 41, 42, 43, 44.

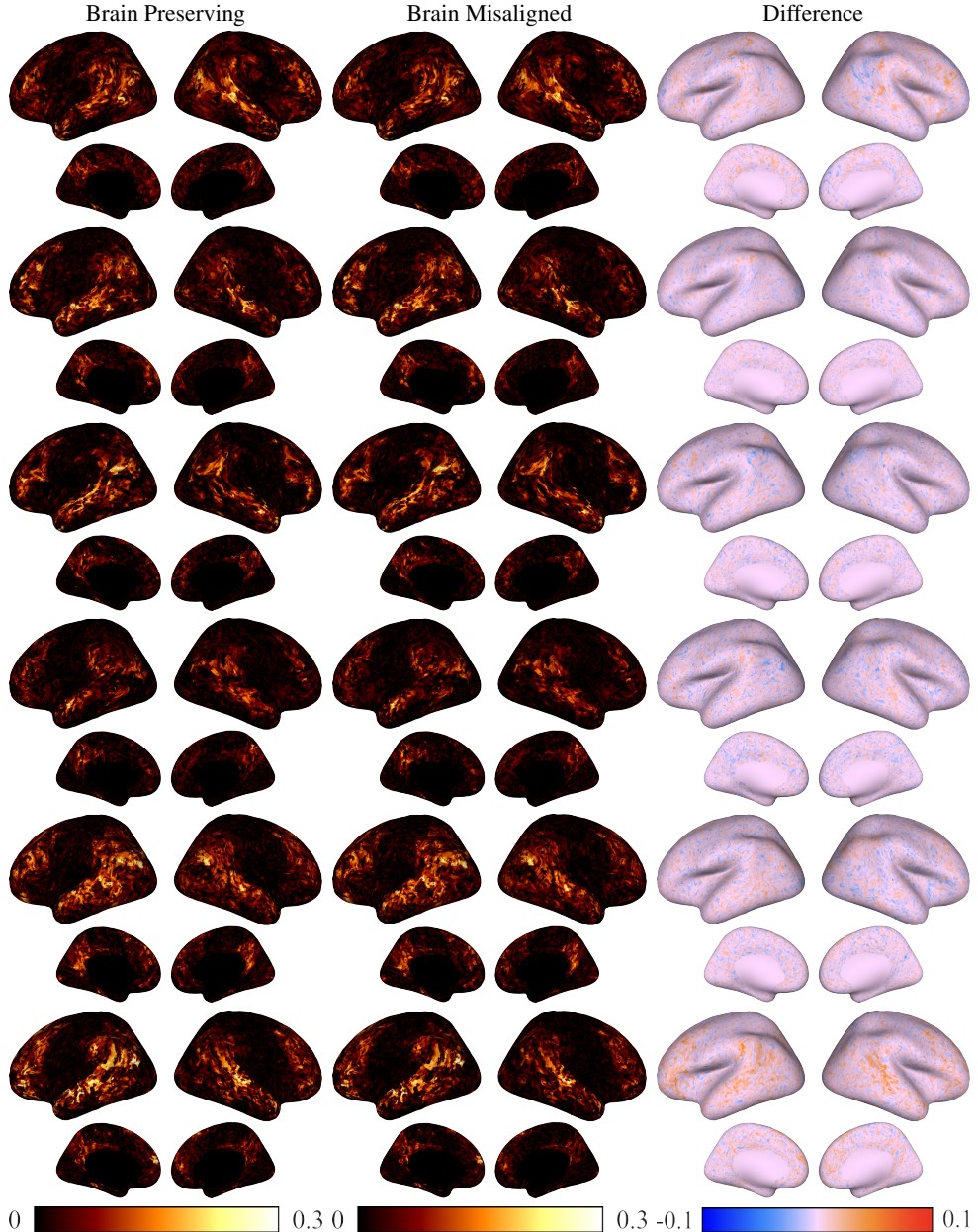

Figure 36: Performances of GPT2-based Brain Misaligned and Brain Preserving models on the Moth Radio Hour dataset at the brain alignment task. Brain plots show voxel-wise Pearson correlations between model activations and brain responses for each subject. The left column displays results for the Brain Preserving model, the center column for the Brain Misaligned model, and the right column shows their difference (Preserving minus Misaligned). Warmer colors indicate stronger alignment with brain activity. These results illustrate the distribution of brain alignment across subjects and highlight areas where brain misalignment has effects.

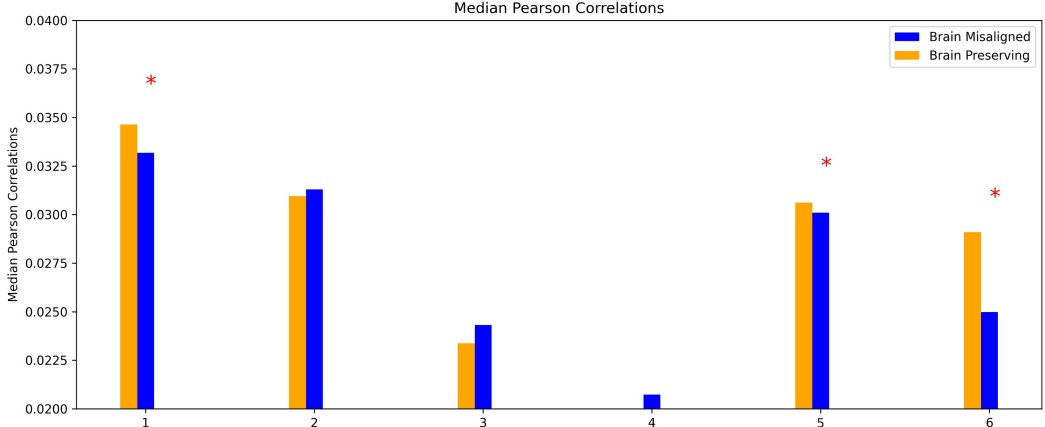

Figure 37: Median Pearson correlation for GPT2-based models on Moth Radio Hour dataset for each participant. Brain Misaligned models perform significantly worse than Brain Preserving models for 3 subjects ($p < 0.05$, indicated by * and assessed using the Wilcoxon signed-rank test).

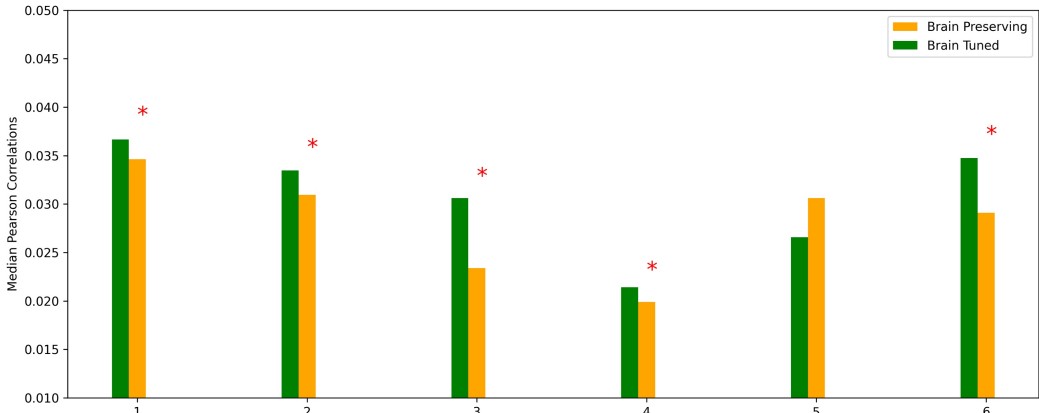

Figure 38: Median Pearson correlation for GPT2-based models on Moth Radio Hour dataset for each participant. Brain Preserving models perform significantly worse than Brain Tuned models for six subjects ($p < 0.05$, indicated by * and assessed using the Wilcoxon signed-rank test).

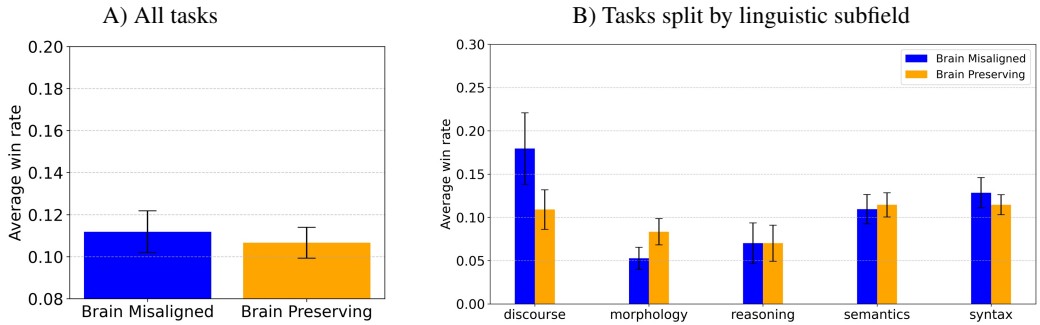

Figure 39: Average win rate and standard error of the GPT2-based Brain Misaligned and Brain Preserving models on the Moth Radio Hour dataset across participants and tasks (Left) and across different linguistic subfields (Right). The win rate indicates how often each model outperforms its counterpart across tasks and participants. The Brain Preserving model shows a higher win rate (Right) in the semantics and morphology subfield (although Wilcoxon signed-rank test with Holm-Bonferroni correction reveal no significance), suggesting that removing brain alignment particularly affects semantics and morphology tasks.

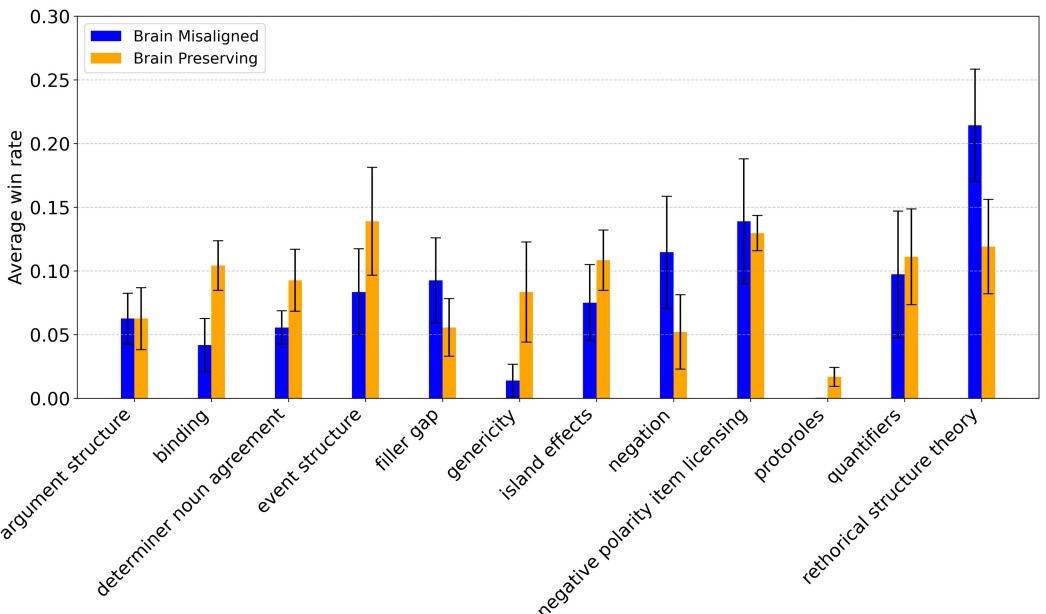

Figure 40: Average win rate with standard error across various linguistic phenomena for the GPT2-based Brain Misaligned and Brain Preserving models on the Moth Radio Hour dataset. Each bar represents the average win rate for a specific linguistic phenomenon, with error bars indicating standard error. Brain Preserving models tend to outperform Brain Misaligned models, particularly in categories such as genericity, event structure and binding. Some concrete examples of the linguistic tasks are provided in the Table 2.

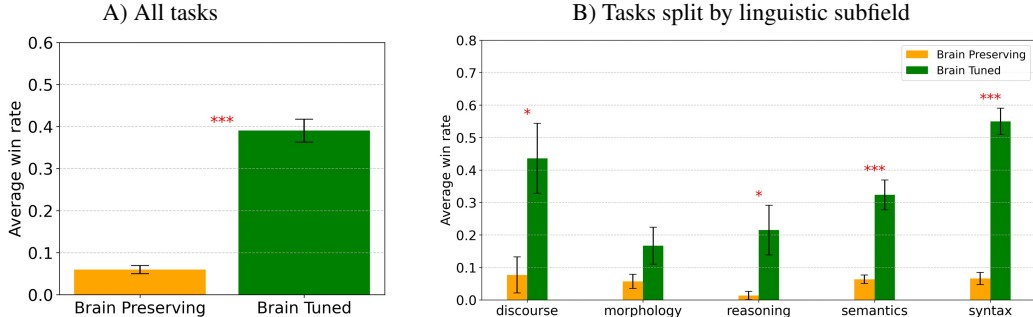

Figure 41: Average win rate and standard error of the GPT2-based Brain Preserving and Brain Tuned models on the Moth Radio Hour dataset across participants and tasks (Left) and across different linguistic subfields (Right). The win rate indicates how often each model outperforms its counterpart across tasks and participants. The Brain Tuned model significantly outperforms the Brain Preserving model ($p < 0.001$, indicated by ***), as assessed using a Wilcoxon signed-rank test (Left). This result suggests that improving brain alignment positively influences linguistic competence. The Brain Tuned model shows a higher win rate in the syntax, semantics, reasoning, morphology and discourse subfield (Right) and significantly higher for syntax, semantics, reasoning and discourse subfields ($p < 0.05$, Wilcoxon signed-rank test with Holm-Bonferroni correction), suggesting that improving brain alignment affects those linguistic subfields.

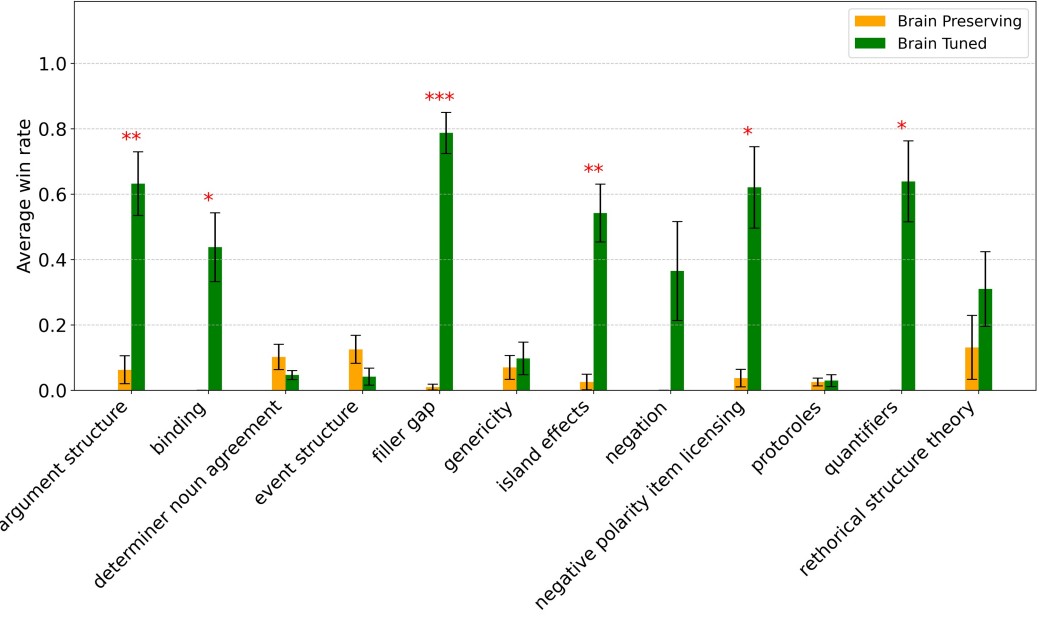

Figure 42: Average win rate with standard error across various linguistic phenomena for the GPT2-based Brain Preserving and Brain Tuned models on the Moth Radio Hour dataset. Each bar represents the average win rate for a specific linguistic phenomenon, with error bars indicating standard error. Some concrete examples of the linguistic tasks are provided in the Table 2.

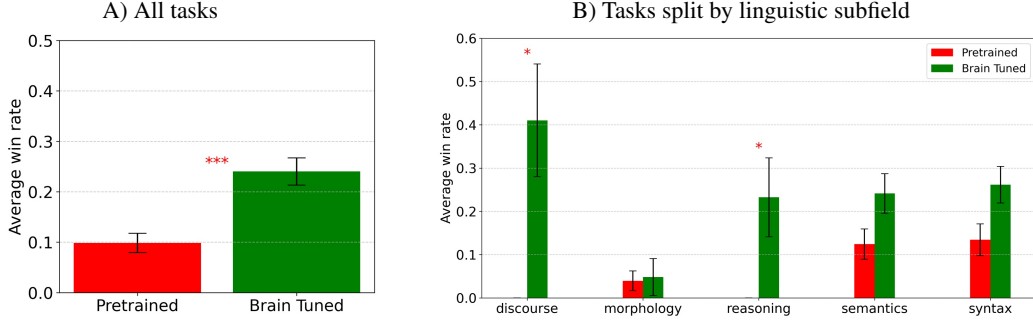

Figure 43: Average win rate and standard error of the GPT2-based Brain Tuned and Pretrained models on the Moth Radio Hour dataset across participants and tasks (Left) and across different linguistic subfields (Right). The win rate indicates how often each model outperforms its counterpart across tasks and participants. The Brain Tuned model significantly outperforms the Pretrained model ($p < 0.001$, indicated by ***), as assessed using a Wilcoxon signed-rank test (Left). This result suggests that improving brain alignment positively influences linguistic competence. The Brain Tuned model shows a higher win rate in the syntax, semantics, reasoning, morphology and discourse subfield (Right) and significantly higher for reasoning and discourse subfields ($p < 0.05$, Wilcoxon signed-rank test with Holm-Bonferroni correction), suggesting that improving brain alignment affects those linguistic subfields.

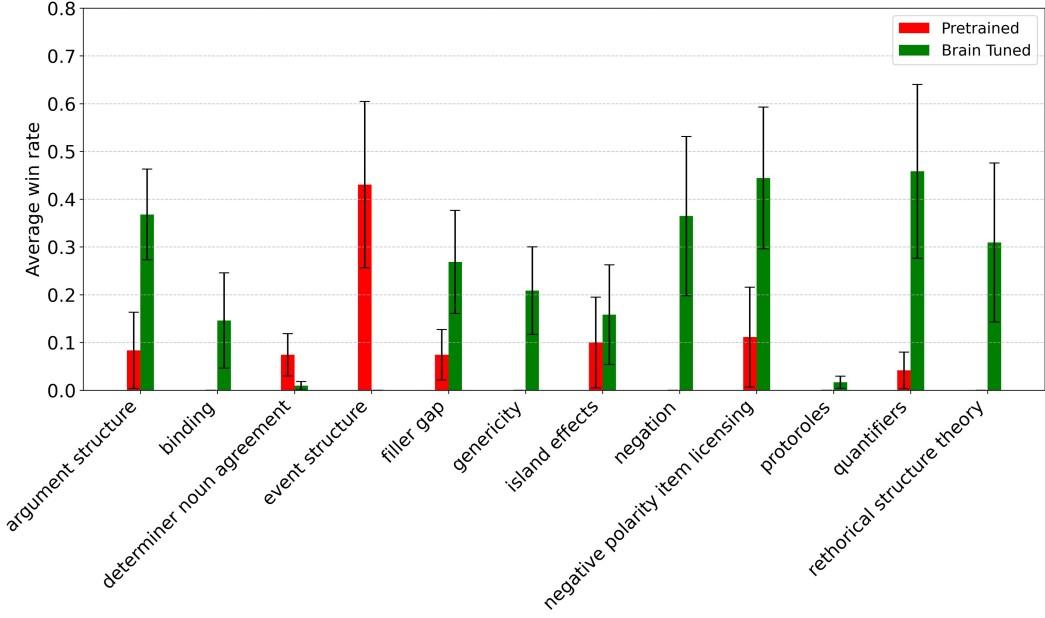

Figure 44: Average win rate with standard error across various linguistic phenomena for the GPT2-based Brain Tuned and Pretrained models on the Moth Radio Hour dataset. Each bar represents the average win rate for a specific linguistic phenomenon, with error bars indicating standard error. Some concrete examples of the linguistic tasks are provided in the Table 2.

## H   LLAMA MISALIGNMENT ON HARRY POTTER DATASET

We report the brain alignment results for Brain Misaligned and Brain Preserving trained with data from each participant in Figure 45, as well as a quantitative summary in Figure 46. Figure 47 report the quantitative summary for brain alignment for the Brain Tuned model compared to Brain Preserving model. Results for the Holmes benchmark for all the comparisons are reported in Figure 48, 49, 50, 51, 52, 53.

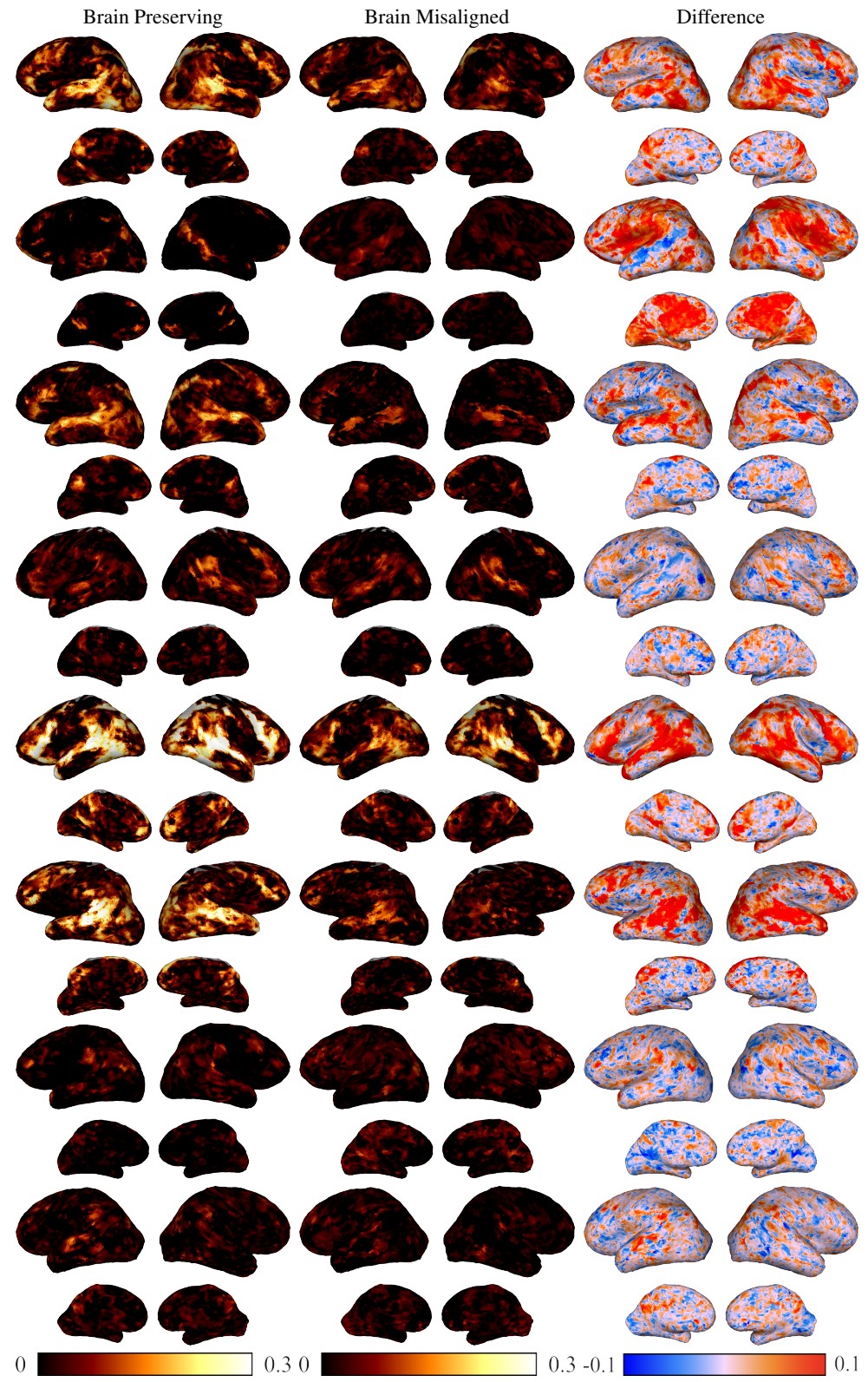

Figure 45: Performances of Llama-based Brain Misaligned and Brain Preserving models on the Harry Potter dataset at the brain alignment task. Brain plots show voxel-wise Pearson correlations between model activations and brain responses for each subject. The left column displays results for the Brain Preserving model, the center column for the Brain Misaligned model, and the right column shows their difference (Preserving minus Misaligned). Warmer colors indicate stronger alignment with brain activity. These results illustrate the distribution of brain alignment across subjects and highlight areas where brain misalignment has effects.

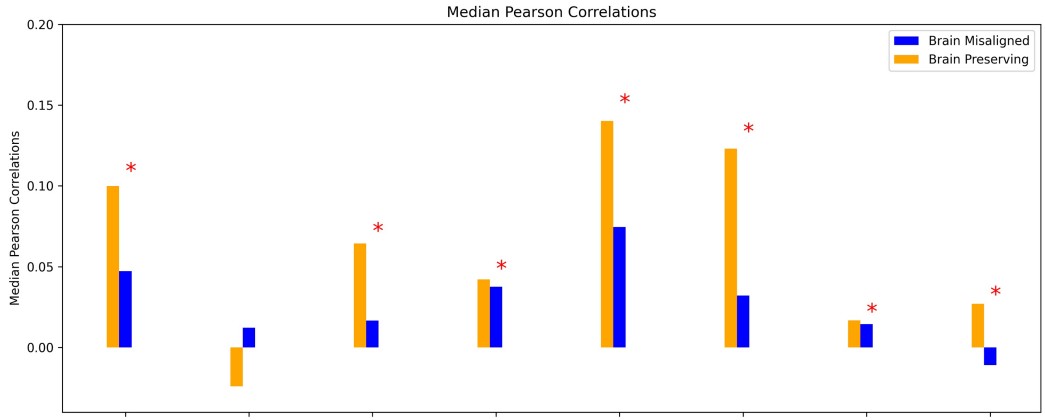

Figure 46: Median Pearson correlation for Llama-based models on the Harry Potter dataset for each participant. Brain Misaligned models perform significantly worse than Brain Preserving models for seven subjects ($p < 0.05$, indicated by * and assessed using the Wilcoxon signed-rank test).

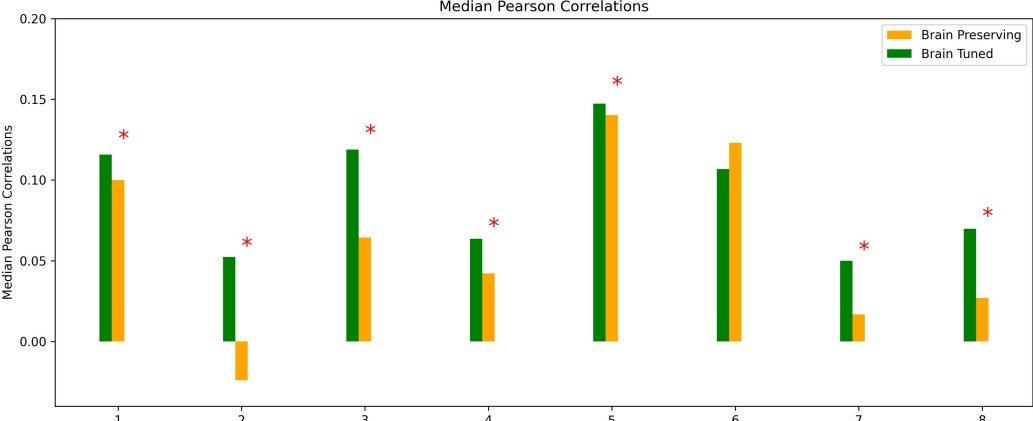

Figure 47: Median Pearson correlation for Llama-based models on the Harry Potter dataset for each participant. Brain Preserving models perform significantly worse than Brain Tuned models for seven subjects ($p < 0.05$, indicated by * and assessed using the Wilcoxon signed-rank test).

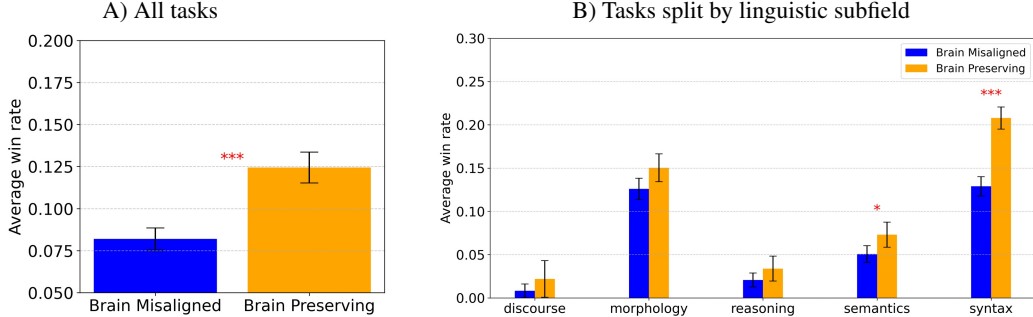

Figure 48: Average win rate and standard error of the Llama-based Brain Misaligned and Brain Preserving models on the Harry Potter dataset across participants and tasks (Left) and across different linguistic subfields (Right). The win rate indicates how often each model outperforms its counterpart across tasks and participants. The Brain Preserving model significantly outperforms the Brain Misaligned model ($p < 0.001$, indicated by ***), as assessed using a Wilcoxon signed-rank test (Left). This result suggests that removing brain alignment negatively influences linguistic competence. The Brain Preserving model shows a higher win rate in the syntax, semantics, reasoning, morphology and discourse subfield (Right) and significantly higher for syntax and semantics subfields ($p < 0.05$, Wilcoxon signed-rank test with Holm-Bonferroni correction), suggesting that removing brain alignment particularly affects syntax and semantic tasks.

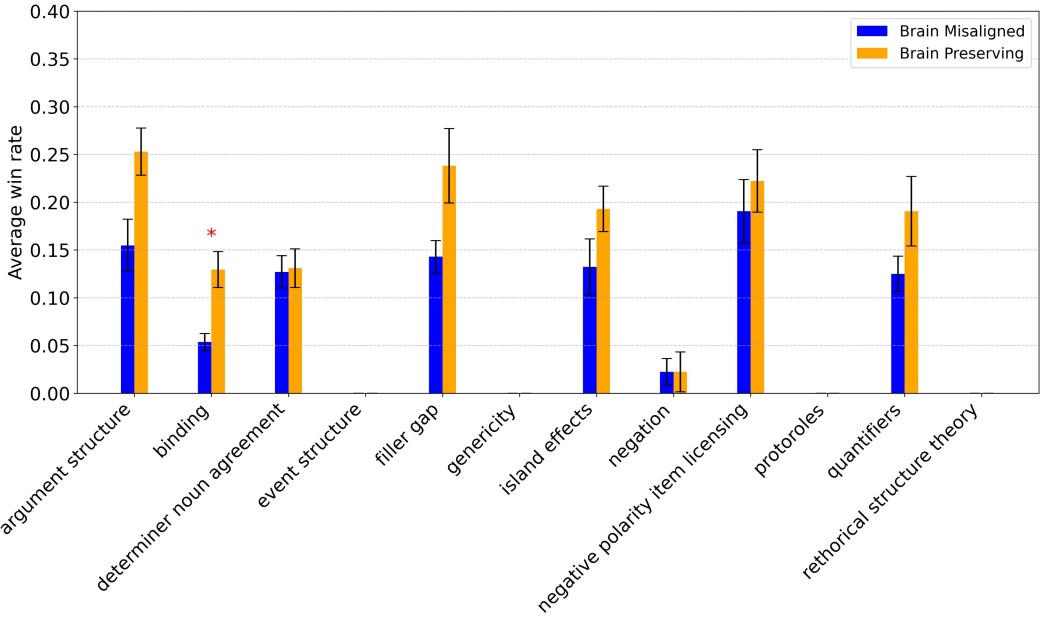

Figure 49: Average win rate with standard error across various linguistic phenomena for the Llama-based Brain Misaligned and Brain Preserving models on the Harry Potter dataset. Each bar represents the average win rate for a specific linguistic phenomenon, with error bars indicating standard error. Some concrete examples of the linguistic tasks are provided in the Table 2.

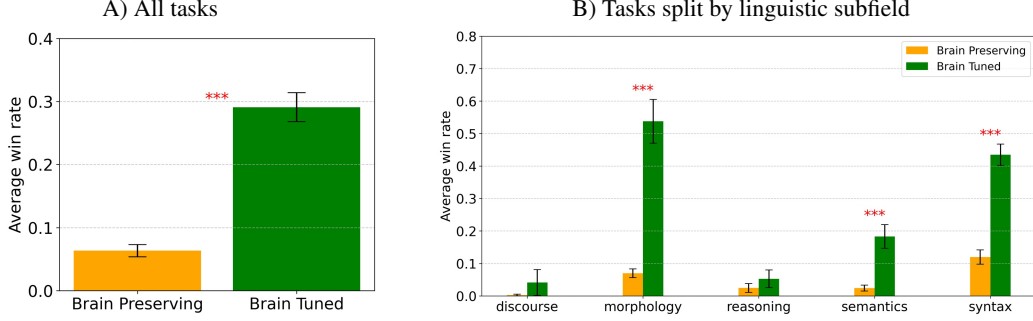

Figure 50: Average win rate and standard error of the Llama-based Brain Preserving and Brain Tuned models on the Harry Potter dataset across participants and tasks (Left) and across different linguistic subfields (Right). The win rate indicates how often each model outperforms its counterpart across tasks and participants. The Brain Tuned model significantly outperforms the Brain Preserving model ($p < 0.001$, indicated by ***), as assessed using a Wilcoxon signed-rank test (Left). This result suggests that improving brain alignment positively influences linguistic competence. The Brain Tuned model shows a higher win rate in the syntax, semantics, reasoning, morphology and discourse subfield (Right) and significantly higher for syntax, semantics and morphology subfields ($p < 0.05$, Wilcoxon signed-rank test with Holm-Bonferroni correction), suggesting that improving brain alignment affects all linguistic subfields.

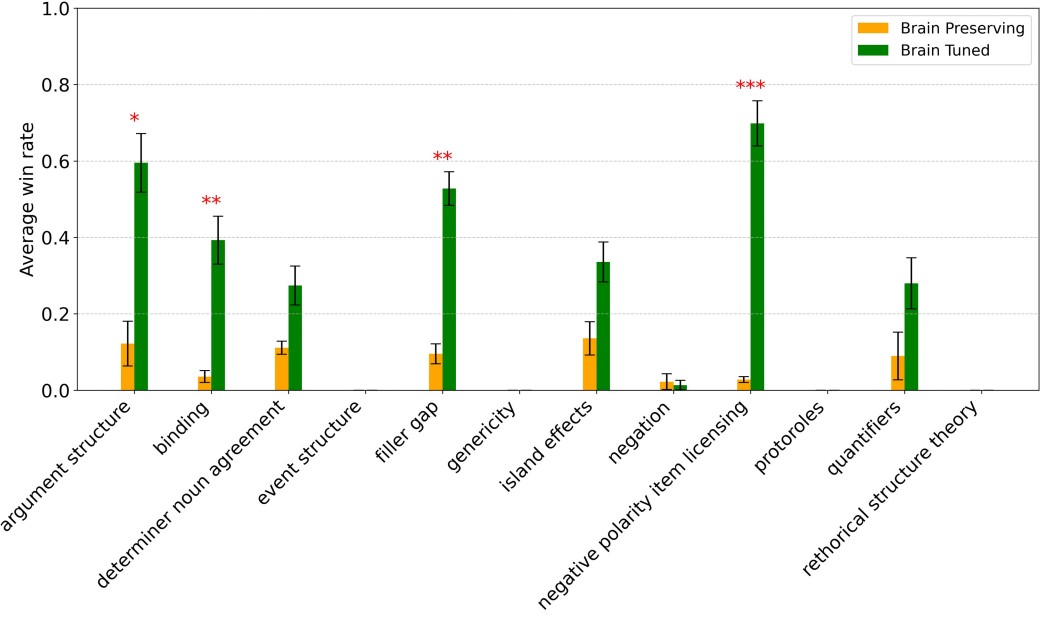

Figure 51: Average win rate with standard error across various linguistic phenomena for the Llama-based Brain Preserving and Brain Tuned models on the Harry Potter dataset. Each bar represents the average win rate for a specific linguistic phenomenon, with error bars indicating standard error. Some concrete examples of the linguistic tasks are provided in the Table 2.

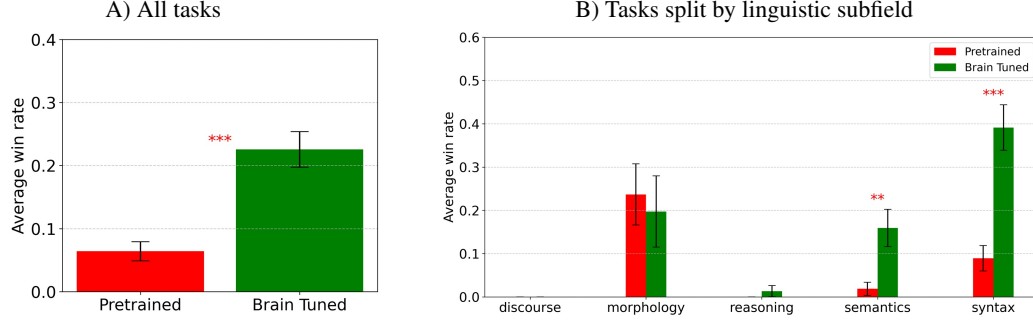

Figure 52: Average win rate and standard error of the Llama-based Brain Tuned and Pretrained models on the Harry Potter dataset across participants and tasks (Left) and across different linguistic subfields (Right). The win rate indicates how often each model outperforms its counterpart across tasks and participants. The Brain Tuned model significantly outperforms the Pretrained model ($p < 0.001$, indicated by ***), as assessed using a Wilcoxon signed-rank test (Left). This result suggests that improving brain alignment positively influences linguistic competence. The Brain Tuned model shows a higher win rate in the syntax, semantics and reasoning subfields (Right) and significantly higher for semantics and syntax subfields ($p < 0.05$, Wilcoxon signed-rank test with Holm-Bonferroni correction), suggesting that improving brain alignment affects those linguistic subfields.

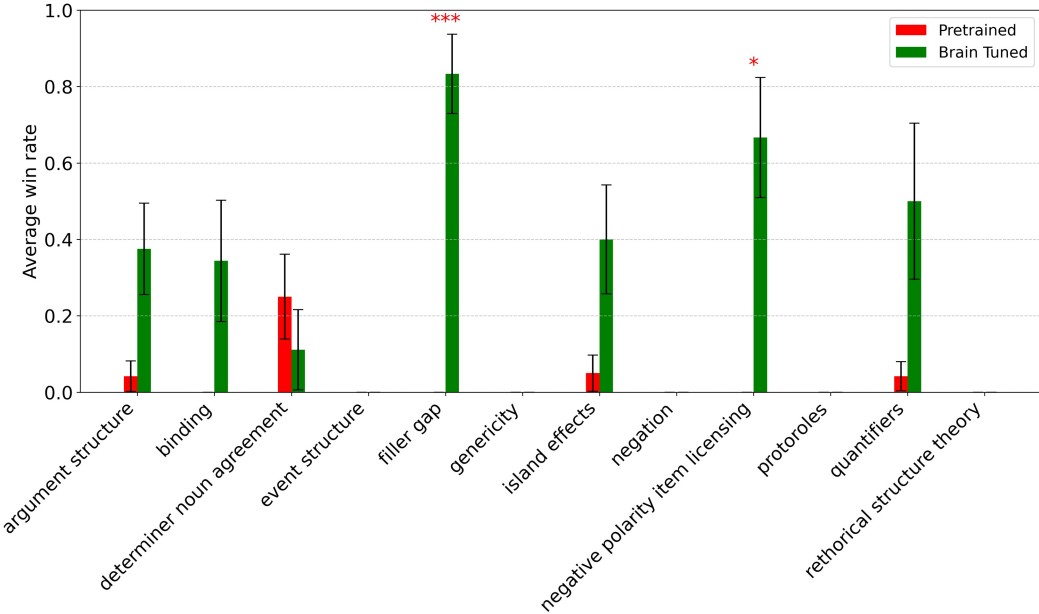

Figure 53: Average win rate with standard error across various linguistic phenomena for the Llama-based Brain Tuned and Pretrained models on the Harry Potter dataset. Each bar represents the average win rate for a specific linguistic phenomenon, with error bars indicating standard error. Some concrete examples of the linguistic tasks are provided in the Table 2.

# I  LLAMA MISALIGNMENT ON MOTH RADIO HOUR DATASET

We report the brain alignment results for Brain Misaligned and Brain Preserving trained with data from each participant in Figure 54, as well as a quantitative summary in Figure 55. Figure 56 report the quantitative summary for brain alignment for the Brain Tuned model compared to Brain Preserving model. Results for the Holmes benchmark for all the comparisons are reported in Figure 57, 58, 59, 60, 61, 62.

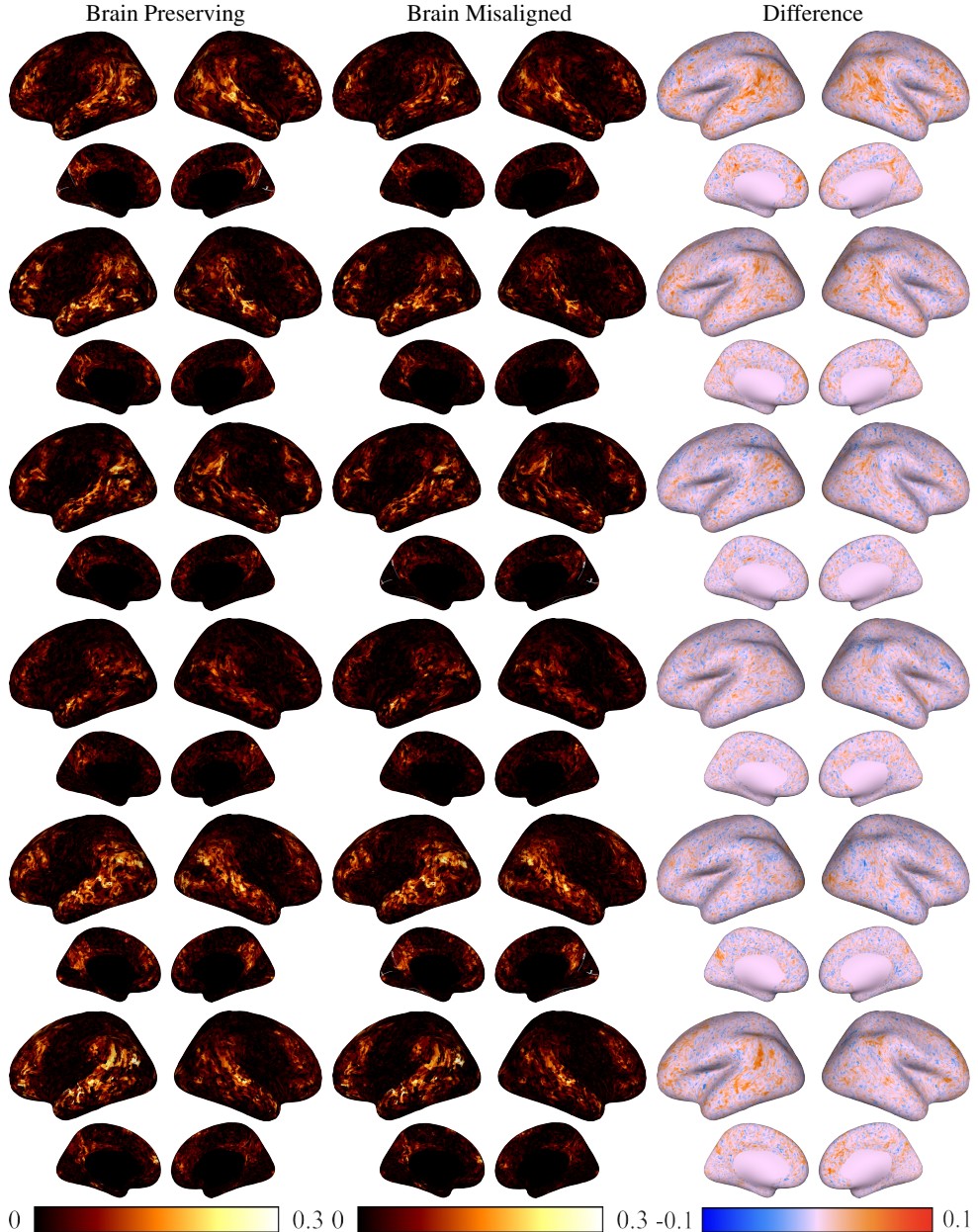

Figure 54: Performances of Llama-based Brain Misaligned and Brain Preserving models on the Moth Radio Hour dataset at the brain alignment task. Brain plots show voxel-wise Pearson correlations between model activations and brain responses for each subject. The left column displays results for the Brain Preserving model, the center column for the Brain Misaligned model, and the right column shows their difference (Preserving minus Misaligned). Warmer colors indicate stronger alignment with brain activity. These results illustrate the distribution of brain alignment across subjects and highlight areas where brain misalignment has effects.

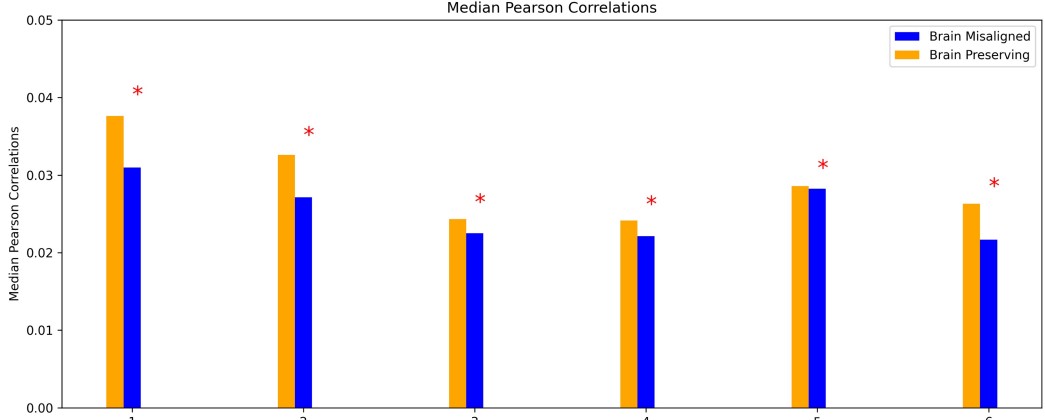

Figure 55: Median Pearson correlation for Llama-based models on the Moth Radio Hour dataset for each participant. Brain Misaligned models perform significantly worse than Brain Preserving models for eight subjects ($p < 0.05$, indicated by * and assessed using the Wilcoxon signed-rank test).

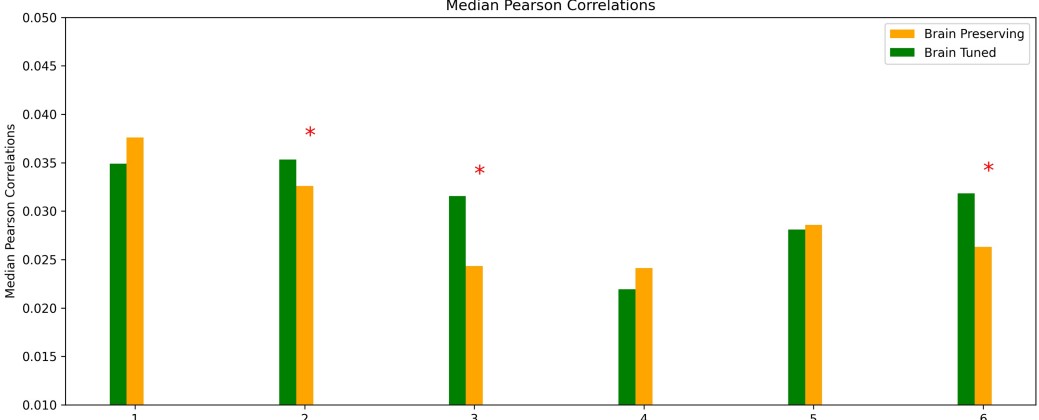

Figure 56: Median Pearson correlation for Llama-based models on the Harry Potter dataset for each participant. Brain Preserving models perform significantly worse than Brain Tuned models for three subjects ($p < 0.05$, indicated by * and assessed using the Wilcoxon signed-rank test).

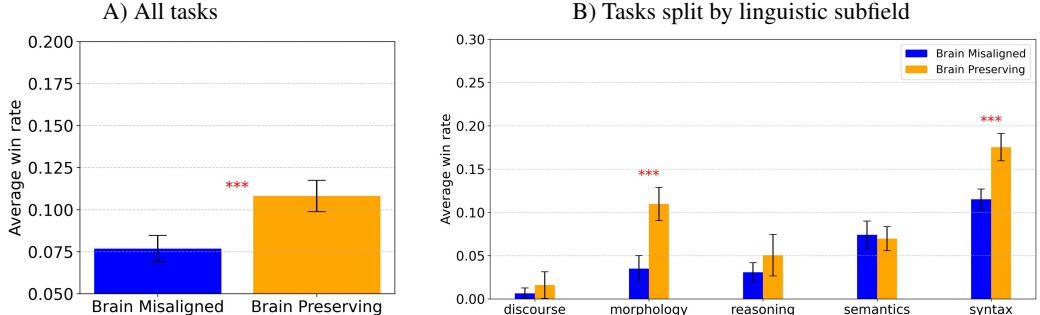

Figure 57: Average win rate and standard error of the Llama-based Brain Misaligned and Brain Preserving models on the Moth Radio Hour dataset across participants and tasks (Left) and across different linguistic subfields (Right). The win rate indicates how often each model outperforms its counterpart across tasks and participants. The Brain Preserving model significantly outperforms the Brain Misaligned model ($p < 0.001$, indicated by ***), as assessed using a Wilcoxon signed-rank test (Left). This result suggests that removing brain alignment negatively influences linguistic competence. The Brain Preserving model shows a higher win rate in the syntax, reasoning, morphology and discourse subfield (Right) and significantly higher for syntax and morphology subfields ($p < 0.05$, Wilcoxon signed-rank test with Holm-Bonferroni correction), suggesting that removing brain alignment particularly affects syntax and morphology tasks.

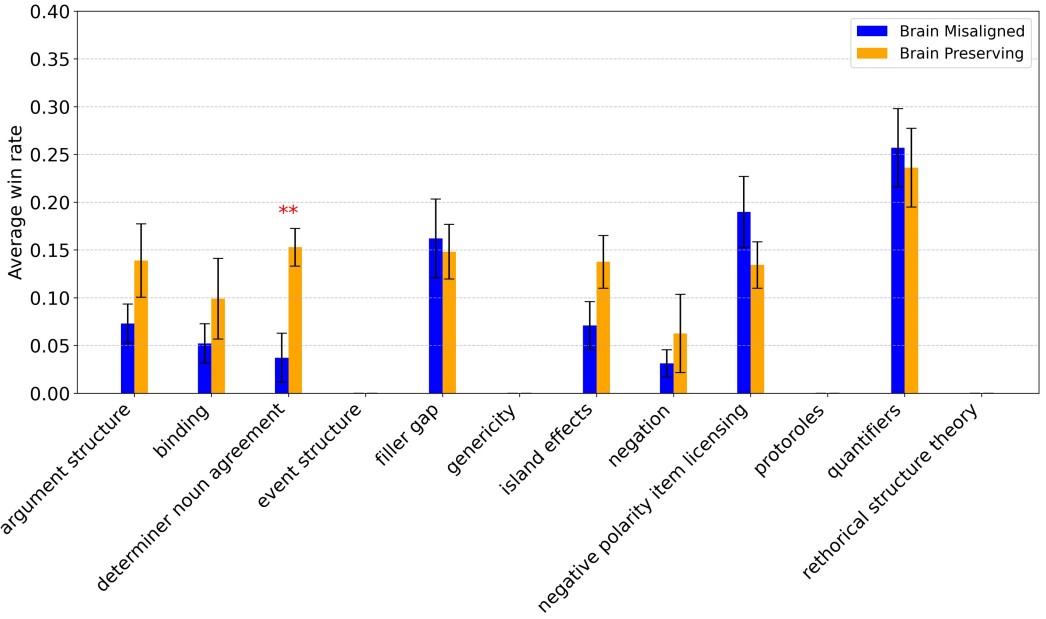

Figure 58: Average win rate with standard error across various linguistic phenomena for the Llama-based Brain Misaligned and Brain Preserving models on the Moth Radio Hour dataset. Each bar represents the average win rate for a specific linguistic phenomenon, with error bars indicating standard error. Some concrete examples of the linguistic tasks are provided in the Table 2.

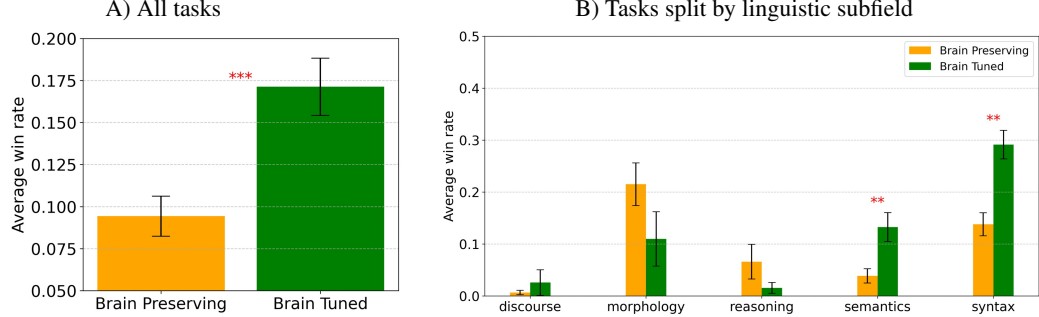

Figure 59: Average win rate and standard error of the Llama-based Brain Preserving and Brain Tuned models on the Moth Radio Hour dataset across participants and tasks (Left) and across different linguistic subfields (Right). The win rate indicates how often each model outperforms its counterpart across tasks and participants. The Brain Tuned model significantly outperforms the Brain Preserving model ($p < 0.001$, indicated by ***), as assessed using a Wilcoxon signed-rank test (Left). This result suggests that improving brain alignment positively influences linguistic competence. The Brain Tuned model shows a higher win rate in the syntax, semantics and discourse subfields (Right) and significantly higher for syntax and semantics subfields ($p < 0.05$, Wilcoxon signed-rank test with Holm-Bonferroni correction), suggesting that improving brain alignment affects those linguistic subfields.

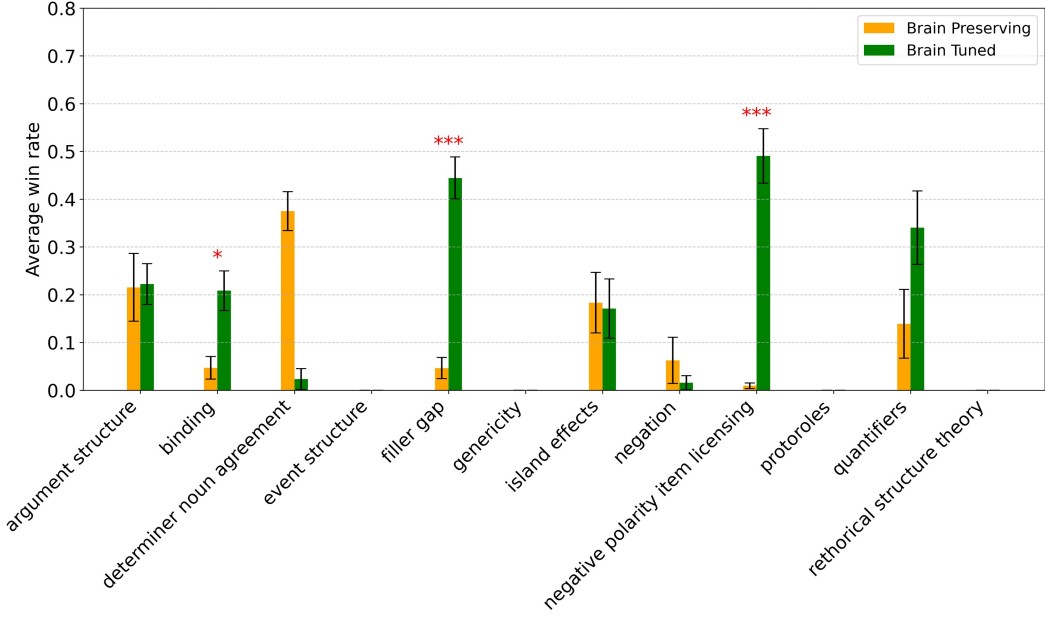

Figure 60: Average win rate with standard error across various linguistic phenomena for the Llama-based Brain Preserving and Brain Tuned models on the Moth Radio Hour dataset. Each bar represents the average win rate for a specific linguistic phenomenon, with error bars indicating standard error. Some concrete examples of the linguistic tasks are provided in the Table 2.

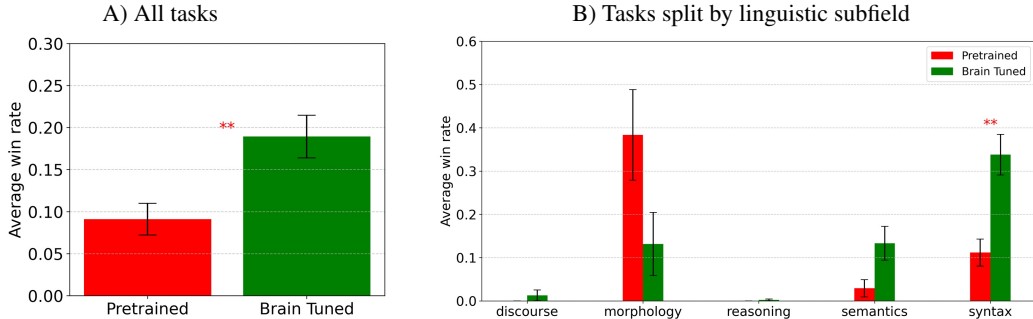

Figure 61: Average win rate and standard error of the Llama-based Brain Tuned and Pretrained models on the Moth Radio Hour dataset across participants and tasks (Left) and across different linguistic subfields (Right). The win rate indicates how often each model outperforms its counterpart across tasks and participants. The Brain Tuned model significantly outperforms the Pretrained model ($p < 0.01$, indicated by **), as assessed using a Wilcoxon signed-rank test (Left). This result suggests that improving brain alignment positively influences linguistic competence. The Brain Tuned model shows a higher win rate in the syntax, semantics, reasoning and discourse subfield (Right) and significantly higher for syntax ($p < 0.05$, Wilcoxon signed-rank test with Holm-Bonferroni correction), suggesting that improving brain alignment affects this linguistic subfield.

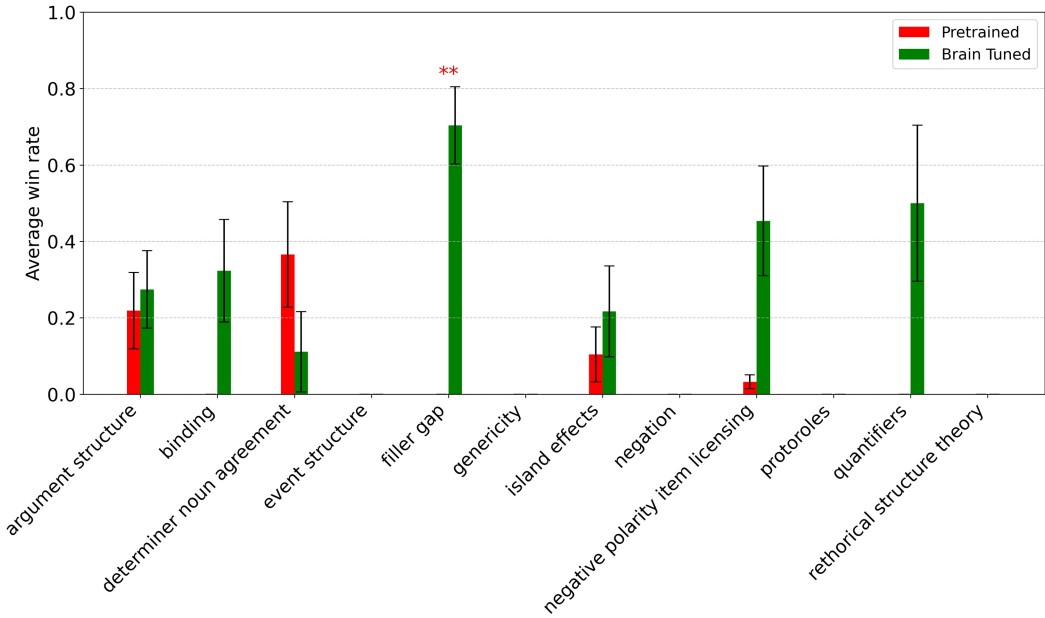

Figure 62: Average win rate with standard error across various linguistic phenomena for the Llama-based Brain Tuned and Pretrained models on the Moth Radio Hour dataset. Each bar represents the average win rate for a specific linguistic phenomenon, with error bars indicating standard error. Some concrete examples of the linguistic tasks are provided in the Table 2.

.

# J    AVERAGED COMPARISONS: ADDITIONAL RESULTS

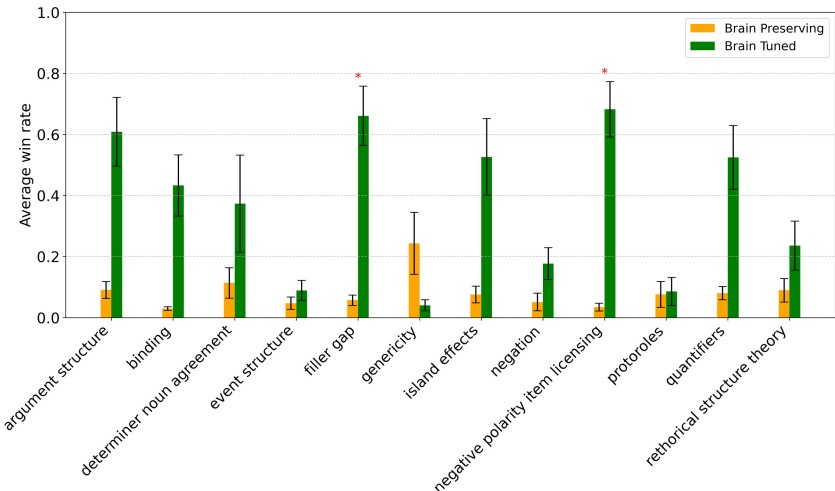

Figure 63: Averaged win rate with standard error for all model and dataset combinations across various linguistic phenomena of the Brain Tuned and Brain Preserving models. Each bar represents the average win rate for a specific linguistic phenomenon, with error bars indicating standard error. Brain Tuned models tend to outperform Brain Preserving models, particularly in categories such as `filler gap` and `negative polarity item licensing`. Some concrete examples of the linguistic tasks are provided in the Table 2.

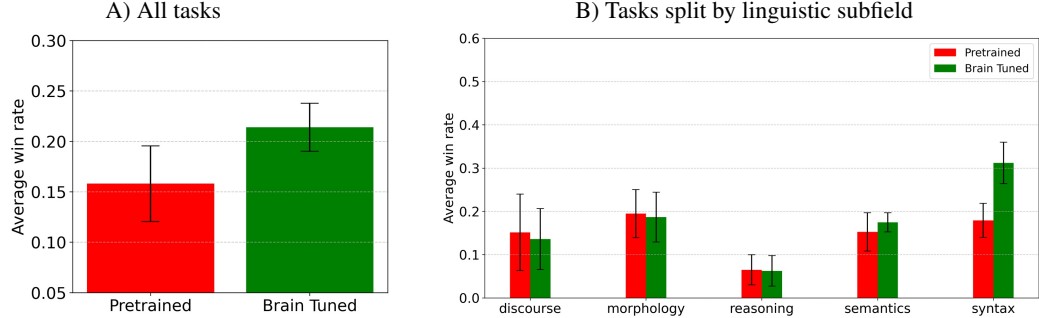

Figure 64: Averaged win rate and standard error for all the model and dataset combinations of the Brain Tuned and Pretrained models across tasks (Left) and across different linguistic subfields (Right). The win rate indicates how often each model outperforms its counterpart across tasks and participants. The Brain Tuned models outperform the Pretrained models (Left). This result suggests that train to align with brain recording leads to improvement in linguistic competence. The Brain Tuned model shows a higher win rate in the syntax and semantics subfields (Right) (although Wilcoxon signed-rank test with Holm-Bonferroni correction reveal no significance), suggesting that removing brain alignment particularly affects those tasks.

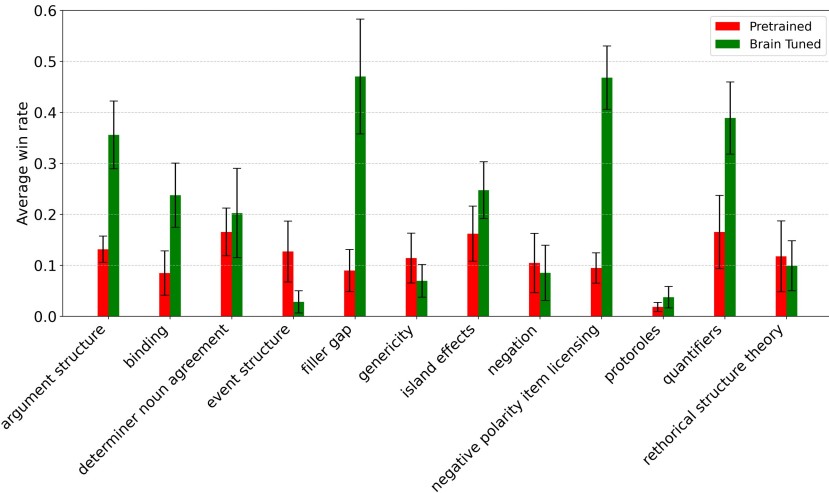

Figure 65: Averaged win rate and standard error for all the model and dataset combinations of the Brain Tuned and Pretrained models across different linguistic phenomena (Right). Each bar represents the average win rate for a specific linguistic phenomenon, with error bars indicating standard error. Brain Tuned models tend to outperform Pretrained models. Some concrete examples of the linguistic tasks are provided in the Table 2.

