# OpenReview forum: "When Language Models Lose Their Mind: The Consequences of Brain Misalignment"
_ICLR.cc/2026/Conference — ICLR 2026 Poster_

### Official Review · Reviewer_Fhx9 · 2025-10-16

**Soundness:** 2
**Presentation:** 2
**Contribution:** 2
**Rating:** 2
**Confidence:** 4

**Summary:**

The authors introduce a novel methodology to create "brain-misaligned" models by fine-tuning pretrained LLMs (BERT and GPT-2) with a dual objective: maintaining language modeling performance while actively training the model's representations to be poor predictors of human fMRI activity recorded during reading. This is achieved using an adversarial setup with a gradient reversal layer. As a control, "brain-preserving" models are trained using the same procedure but with permuted fMRI data, isolating the effect of genuine brain-stimulus alignment. The linguistic competence of these misaligned and preserving models is then comprehensively evaluated on over 200 downstream tasks. The central finding is that deliberately reducing brain alignment significantly impairs a model's performance on these linguistic tasks, particularly in the domains of semantics, syntax, and reasoning.

**Strengths:**

The creation of "brain-misaligned" models is a clever and powerful technique. Using a gradient reversal layer to explicitly penalize brain predictability, while simultaneously preserving language modeling ability, allows the authors to isolate the variable of interest in a way that correlational studies cannot. The "brain preserving" model serves as a well-thought-out control, accounting for potential confounding effects of the fine-tuning procedure itself.

**Weaknesses:**

1. The performance drop is clear and significant for BERT models, but the effect is weaker and only marginally significant for GPT-2 on the Harry Potter dataset. Furthermore, the authors note that for GPT-2 on the Moth Radio Hour dataset, "results are not consistent due to the weaker effect of brain removal." This variability undermines the universality of the core claim. The paper would be much stronger if it could offer a hypothesis or analysis explaining why this discrepancy exists.

2. The authors fail to conduct experiments on the Narratives dataset, which is much larger, contains more subjects, and has been widely adopted in many previous works. Additional experiments on this dataset will provide more reliable insights.

3. The authors only investigate bert and gpt2, which are out-of-date models. Many recent studies in brain-language model alignment use llama models. I think choosing llama models are especially important when you want to investigate the performance of Brain misaligned models.

4. The paper would be far more transparent and impactful if it also reported absolute performance metrics across task categories instead of "win rate". Without this, it is difficult to judge the practical importance of the observed effect.

**Questions:**

na

---

> ### Author Response · Authors · 2025-11-23
>
> We thank the reviewer for the thorough evaluation of our work. We have updated the paper with the requested edits, including the addition of the suggested citations and the correction of typos. We address the specific concerns and questions below. Please also see the General Response, which contains additional relevant information and analysis.
> - [Different Results for GPT-2] The additional experiments on Llama mentioned in the General Response suggest that the observed discrepancies are likely not due to the difference in architecture (i.e., decoder-only), as Llama behaves similarly to BERT. Investigating whether this pattern persists in other models of the GPT-2 family is indeed an interesting next step. Additionally, in the General Response, we discuss new experiments on Brain Tuning (i.e., improving brain alignment instead of decreasing it). These results point in a consistent direction across all models: it appears that while it was difficult to remove brain alignment in GPT-2, it was feasible to increase it, which resulted in positive downstream effects on linguistic competence.
> - [Additional datasets] We agree that investigating additional datasets could strengthen the results. Crucially, our models are trained on a per-subject basis, which necessitates a large amount of data per participant. Although the Narratives dataset contains data for 345 participants, the stimuli vary across subjects. The longest story is approximately 2,000 TRs, which is comparable to the Harry Potter dataset but only half the length of the Moth Radio Hour subset used in our study.
> - [Additional models] We agree that validating our hypothesis on recent models is crucial. We have updated the paper to include experiments with Llama-3.2-1B, as discussed in the General Response. These new results reinforce our core claims, showing consistent patterns across model generations.
> - [Effect size and win rate] Since our benchmark encompasses multiple task types using different evaluation metrics, we selected the win rate to provide a unified measure. We acknowledge that a "standard" win rate can mask small differences. For this reason, we report an "adjusted" win rate that assigns a "win" to a model only for datasets where the difference reaches statistical significance (across seeds), rather than simply outperforming the other (see lines 262-269 in the updated paper). Consequently, the win rates do not sum to one, as there are tasks where statistical significance is not reached.
>
> We hope our responses have been helpful in addressing the reviewer’s questions. If there are any further issues, we would be pleased to provide additional clarification.

---

### Official Review · Reviewer_7k7B · 2025-10-31

**Soundness:** 2
**Presentation:** 3
**Contribution:** 3
**Rating:** 4
**Confidence:** 3

**Summary:**

The authors introduce Brain Misaligned models that are adversarially fine-tuned, with a gradient-reversal head trained to anti-predict fMRI, to decrease brain predictivity while preserving standard LM loss, and Brain Preserving controls trained identically except that fMRI targets are permuted so that adversarial updates cannot meaningfully target the true mapping. The setup uses BERT-base and GPT-2-small, fMRI from Harry Potter reading and Moth Radio Hour stories, and evaluates language modeling, brain predictivity, and downstream linguistic competence via Holmes/FlashHolmes (>200 probing datasets spanning semantics, syntax, morphology, discourse, reasoning). Core findings: (i) the misaligned models reliably reduce voxel-wise brain predictivity in language ROIs without degrading LM loss; (ii) across participants and seeds, misalignment lowers win-rates on the Holmes benchmark overall, with especially robust drops in semantics (and often syntax/reasoning). The pattern replicates across model/dataset pairs, with clearest effects for BERT; GPT-2 shows similar trends with weaker significance in one setting. The paper argues this is causal evidence that brain-aligned information supports linguistic competence.

**Strengths:**

- The paper is conceptually original in treating brain–LM alignment as an intervenable property rather than a purely correlational observation. Constructing “brain-misaligned” variants via adversarial fine-tuning and contrasting them with a permutation-based control is a creative way to probe causal relevance.

- Methodologically, the study is careful about a manipulation check (reduced brain predictivity in language ROIs) and about holding general LM performance roughly fixed during selection, which increases confidence that downstream effects are tied to the intervention rather than to obvious capacity loss. Empirically, the evaluation spans a broad set of linguistic competencies (Holmes/FlashHolmes).

- The paper is generally clear and well organized, with helpful schematics and ROI visualizations. It is a pity that most of the experiments/results are added to the appendix and are not integrated into Figures 3 and 4.

**Weaknesses:**

- The framing somewhat overstates what the intervention establishes. Studying the effects of removing alignment does not automatically prove that alignment, per se, is beneficial. Removing brain–LM similarity and observing performance drops suggests that aligned information may correlate with linguistic competence, but it does not directly prove that alignment itself is necessary or causally beneficial. How do we ensure that the suggested manipulation does not act as structured noise or remove semantically relevant information? It would help to clarify whether this intervention reflects targeted “de-alignment” or a broader perturbation of representational geometry. Similarly, the construct of alignment could be made more robust: the paper currently relies on a single linear voxel-prediction metric, but conclusions might vary across similarity frameworks (e.g., RSA, CKA, or Procrustes) and layers. Reporting whether results hold across metrics, or demonstrating metric invariance, would strengthen causal claims and continue the discussion by recent comparisons ([Bo et al., 2025](https://openreview.net/forum?id=JGANEMteY8)). Finally, the naming of the Brain Preserving model is potentially misleading, since permuting fMRI targets does not intuitively preserve alignment; a label such as Permutation-Control would be clearer, and the distinction between misaligned and control models should appear earlier and be included among the stated contributions.

- The experimental design would be stronger with two additional reference points: a zero-shot base model and a classic brain-tuned model (trained to predict brain activity given language stimuli or vice versa) to help disentangle generic fine-tuning effects from alignment-specific ones. Reporting and visualization also need tightening: please specify the exact number of test examples and how test samples are derived for cross-entropy estimation (especially given the ~1,211 images in Harry Potter), clarify whether multiple tokens per TR are aggregated, and whether averaging is per token or per example.

- Some results are fragile and not adequately discussed. GPT-2/Harry Potter shows only a trend (p≈0.055), and Moth/GPT-2 is inconsistent due to weaker removal; discussing why adversarial removal is less effective for autoregressive models or the Moth stimuli would help.

Overall, to broaden context and strengthen claims, I would recommend adding citations and, where feasible, experiments around: (a) Narratives by [Nastase et al., 2021](https://www.nature.com/articles/s41597-021-01033-3) as a widely used naturalistic fMRI resource, which would diversify stimuli and subjects; (b) [Pereira et al. (2018)](https://www.nature.com/articles/s41467-018-03068-4) as a complementary sentence-level dataset; (c) alternative representational similarity metrics (RSA/CKA/Procrustes) given active debate on metric choice; (d) concept-erasure baselines (INLP, RLACE, IGBP/single-projection) to test method-robustness of the causal effect; (e) Report exact effect sizes and confidence intervals for win-rate differences, not only p-values; (f) Clarify whether LM losses truly matched (distributions, not just means) and include masked-LM vs autoregressive comparability caveats; (g) Finally, consider a small positive-control experiment: brain-tuning the same base models (without adversarial reversal) to increase predictivity and checking for competence gains, to bracket the causal relationship in both directions (cf. recent brain-tuning gains in speech LMs). This would nicely support your core result.

**Questions:**

- How layer-localized is the removal? Did you inspect per-layer brain predictivity to see whether misalignment concentrates in earlier vs later layers, and whether the competence drop tracks specific layers/heads?
- If you compute brain-alignment with RSA or CKA (and perhaps ROI-wise CKA), do you still see the same manipulation check and downstream deficits? Any reason to expect metric dependence? A nice work comparing representation similarity frameworks is by [Williams, 2024](https://openreview.net/forum?id=zMdnnFasgC).
- Could you repeat the intervention with INLP or RLACE on frozen representations (or hybrid: INLP on a projection bottleneck during fine-tuning) to test whether the competence drop is method-agnostic?
- Have you tried Narratives by [Nastase et al., 2021](https://www.nature.com/articles/s41597-021-01033-3) (multiple stories, many subjects) or the sentence-level fMRI dataset by [Pereira et al. (2018)](https://www.nature.com/articles/s41467-018-03068-4) to probe genre and task sensitivity of the effect? If not, what are the expected obstacles (timing alignment, TR windows, etc.)?
- Beyond probing, do misaligned models underperform on GLUE/SuperGLUE, coreference, and NLI when controlling for instruction-tuning? If tested, please report; if not, justify relying solely on probing.
- Your permutation control is strong, but adversarial training can still regularize or distort representation geometry in ways unrelated to brain signals. Can you add a no-brain adversarial control (e.g., adversary trained on shuffled text features or random vectors) to bound generic adversarial side-effects?
- Is there a specific reason competence evaluation relies on probing? Holmes is excellent for linguistic phenomena, but do you think adding task benchmarks (e.g., GLUE/SuperGLUE, coreference, QA) or behavioral psycholinguistic tests would help bridge representational competence to end-task behavior?

**Comments:**
- Lines 31–32 lack a reference to a comprehensive prior survey: [Karamolegkou et al., 2023](https://aclanthology.org/2023.findings-acl.618/).
- Lines 076-090: The contributions paragraph reads partly like a summary of results ("We find that the competence drop is especially pronounced…"). Conceptually, contributions should describe what is contributed (e.g., a causal framework for testing brain alignment), not what was found. I suggest rephrasing to highlight the methodological and theoretical advances, moving empirical claims to the Results section.
- Lines 101-103: I would recommend tightening causal language. For example, I am not sure whether you study if brain alignment *is necessary for* maintaining linguistic competence in language models, maybe I would rephrase it as *is implicated in*.
- By Line 101, the reader still does not know that there are two model "types" used: brain misaligned and brain preserving. In the abstract you mention the brain preserving models as "well-matched brain aligned counterparts". It would be helpful to clarify earlier that both are adversarially fine-tuned variants of the base model, differing only in whether the fMRI data are permuted. This distinction is crucial for interpreting the experimental design and for understanding how the "alignment" being tested is manipulated. Brain-preserving models should be added to your contributions.
- Section 3.3.2 (“Brain Preserving Model”) is confusing to me. If the fMRI responses are permuted during training, how can this model be described as “preserving” brain alignment? You mention that you "permute the order of the fMRI images to disrupt the correspondence between stimuli and brain activity", this to me does not suggests brain preserving. Maybe the name or the explanation needs refinement... perhaps "Brain-Control" or "Permutation-Control" would be more transparent? In addition, a zero-shot baseline and a standard brain-fine-tuned model (finetuned to predict brain signal given language stimuli, or vice versa) would provide valuable reference points, as I have noted above.
- Lines 224–228: Please specify the number of test examples. The current phrasing, "For each test example, it is measured the average cross entropy across the randomly masked tokens", leaves unclear the dataset size and sampling process. Given that the Harry Potter fMRI dataset contains only around 1,211 brain images, it would be important to clarify (i) how many text–fMRI pairs were used for testing, (ii) whether multiple tokens per TR were aggregated, and (iii) how the averaging procedure interacts with the limited sample size. This information is essential for assessing the robustness and generalizability of the reported results.
- Lines 336–337: Please correct the cross-references "Appendix C and ??".
- Figures 3 and 4 would benefit from including a bar with baseline model that is neither misaligned nor brain-preserving (e.g., the original pretrained and/or a fine-tuned version). This would allow readers to disentangle whether observed differences arise from misalignment, permutation control, or general pretraining/fine-tuning effects.
- The statistical significance markers (asterisks) in Figures 3 and 4 are difficult to interpret—please ensure consistent notation and a clear legend.
- In Figures 3 and 11, the brain-misaligned models seem to outperform baselines on morphological phenomena. This is an intriguing and unexpected finding that deserves discussion, as it may indicate selective effects of the adversarial objective across linguistic domains.
- Although the abstract claims experiments on two fMRI datasets and two models, the main paper only visualizes results for BERT + Harry Potter. Including GPT-2 and Moth results (even as additional bars in Figures 3 and 4) would greatly strengthen the consistency and interpretability of the findings. The divergence between BERT and GPT-2 patterns otherwise leaves conclusions somewhat underdetermined.

---

> ### Author Response · Authors · 2025-11-23
>
> We thank the reviewer for the thorough evaluation of our work. We have updated the paper with the requested edits, including the addition of the suggested citations and the correction of typos. We address the specific concerns and questions below. Please also see the General Response, which contains additional relevant information and analysis.
> - [Summary figures in the main paper] Following the reviewer's suggestion, we have included figures that summarize results across model-dataset combinations in the updated main paper, while specific results for each experiment are now reported in the Appendix.
> - [Stronger analysis] Regarding the hypothesis that the suggested manipulation might act as structured noise: when applying brain misalignment, it is possible that other confounding factors play a role, which is why we included the Brain Preserving model, which acts as a "control" condition. If the adversarial training itself adds some noise, this would add noise in both the Brain Misaligned and the Brain Preserving models; therefore, in our comparison, we are already controlling for this confounding factor. Regarding alternative representation similarity frameworks, while we agree that further analysis could be done using other metrics, we use single linear voxel-prediction as it is one of the established methods in the literature for encoding brain data. Each metric has pros and cons; for example, CKA/RSA have been shown not to be sensitive to certain invariances (Nanda et al., 2022). Regarding the "Brain Preserving" name, we use this term because the original pretrained model is already aligned to the brain recordings; therefore, removing the signal related to permuted brain activity should not remove the actual brain alignment.
> - [Brain Tuned and Pretrained models] We thank the reviewer for this suggestion, and we believe that this point has indeed strengthened our conclusions. We have now conducted additional experiments on Brain Tuned and Pretrained models. We have updated the paper accordingly and have submitted a General Response to highlight the importance of these new results. Please see the General Response for further discussion. To summarize, the Brain Tuned model results in better performance than the Brain Preserving and Brain Misaligned models in every model and dataset combination, nicely supporting our previous conclusions about the importance of brain alignment. Moreover, for the majority of experimental settings, the Brain Tuned model is also superior to the Pretrained model, highlighting the importance of brain data for fine-tuning and its effects on downstream tasks.
> - [Number of test examples] We reported the details on the Brain Mapping Head in lines 239-254 and in Appendix Section B. To summarize, each dataset is divided into four folds of similar size; each proposed model is trained on three folds and tested on the held-out set. For the text-fMRI pair, we concatenate a context of words belonging to the previous 5 TRs.
> - [Results for GPT-2] To investigate the different results for GPT-2, we added new experiments that are also discussed in the General Response and summarized below. We included a new model, Llama-3.2-1B, to enlarge the set of experiments and to further investigate if another autoregressive model would yield similar results; for that model, we were able to remove brain alignment, and the results are in line with those for BERT. Moreover, adding the Brain Tuned models, as mentioned before, helps in disentangling the importance of brain alignment for GPT-2. It seems that even if we were not able to effectively remove the brain alignment, we were able to improve it via brain tuning, and this led to better results in downstream performance. Future work can study if this phenomenon is specific to the GPT-2 family by investigating additional models.
> - [Additional datasets] Indeed, investigating additional datasets could strengthen the results. However, in our setting, we decided to opt for more naturalistic datasets that contain information beyond the sentence level. Regarding similar naturalistic datasets, our models are trained by subject; therefore, we needed a dataset that contains enough data per subject. Even if Narratives contains data for 345 participants, we do not have the same stimuli for all of them. The longest story is about 2k TRs, which is similar to the Harry Potter dataset and half the TRs of the subset of the Moth Radio Hour used here.
> - [Other removal methods] We agree that having more concept-erasure baselines could broaden the context; however, they can sometimes lead to the removal of relevant features beyond the ones we want to remove (Kumar et al., 2023). Instead, adversarial training does not ensure full removal, but in our case, a significant decrease is sufficient. Furthermore, additional features can be preserved using a loss component designed to maintain linguistic ability, similarly to what is done in (Feder et al., 2022).
>
> (continue)

---

> > ### Author Response · Authors · 2025-11-23
> >
> > - [Localization of the removal] We focused on the last layer as it serves as the interface for the probing classifiers, but we acknowledge that layer-wise analysis is a valuable direction for future work.
> > - [Other linguistic benchmarks] Indeed, testing on more benchmarks could strengthen the results; however, Holmes covers more than 200 tasks and can be used as a framework for comparing linguistic competence across different linguistic subfields. We have updated the paper to include citations to other benchmarks.
> > - [Other Controls] Using random vectors or distinct features would introduce a distributional mismatch compared to fMRI data, acting as a confounding factor that makes results hard to interpret. Our permuted brain data control is specifically designed to be the rigorous counterfactual: it preserves the statistics and distributional properties of the brain signals while destroying only the semantic link. This ensures that any observed effect is strictly due to the removal of brain-aligned information, rather than generic adversarial instability or regularization artifacts.
> >
> > We hope that our clarifications have addressed the reviewer’s concerns. If there are any further issues, we would be pleased to provide additional clarification.

---

> > > ### Comment · Reviewer_7k7B · 2025-11-27
> > > **Acknowledgment of Author Response**
> > >
> > > Thank you for your responses. After careful consideration, I have adjusted my score accordingly.

---

### Official Review · Reviewer_J54j · 2025-10-31

**Soundness:** 3
**Presentation:** 3
**Contribution:** 3
**Rating:** 6
**Confidence:** 5

**Summary:**

The paper trains LLMs to *poorly* predict brain activity (Wehbe et al. 2014 HP and Deniz et al. 2019 MRH fMRI datasets), in order to test the relevance of brain alignment for language task performance.
The main claim is that LLMs misaligned to the brain exhibit poor downstream performance, with which the authors argue that brain alignment is critical for linguistic competence.

**Strengths:**

1. Explicitly training models for brain misalignment is novel as far as I am aware. This is a cool method for causally testing the effect of neural data regularization in both directions.

2. Great control: models trained for misalignment are compared with models under the same training pipeline but with permuted stimulus-response pairs.

3. Methods and results are presented clearly, making the paper easy to follow.

**Weaknesses:**

### 1. Small scale of tested models
The models under consideration (BERT-base and GPT-small) are very small by modern standards, limiting the applicability of claims to the state of the art. In particular, smaller models might lack capacity to retain high linguistic performance despite reduced brain alignment, whereas larger models' performance might be affected by brain-misalignment less strongly or not at all.

### 2. Small effect size
The average win rate between misaligned model and control differs by 2 percent points (0.19 vs 0.21). I further wonder what the actual difference in performance is. Even small performance differences can result in different win rates, but the effect size nonetheless can well be very small.

### 3. Increased alignment of misaligned model in some regions
There is something weird with the training: why are the models trained to be misaligned actually more aligned for a substantial number of voxels (Fig. 2C)? It seems the authors attribute this to such regions not being language-related, but since the model is trained with whole-brain data this shouldn't happen.


### Minor
* L247 typo "more tha**t** 200 datasets"
* L337 missing reference "??"
* Figure 3A: shouldn't the win rate between two models sum up to 1?

**Questions:**

Please see Weaknesses 1-3 in particular. I am willing to increase my score with the evaluation of larger models (e.g., Qwen2.5-72B, Pythia, OLMo), and a quantification of performance deltas (rather than win rate).

I like the misalignment training method! Please describe related work of training models with brain data.

---

> ### Author Response · Authors · 2025-11-23
>
> We thank the reviewer for the thoughtful and constructive comments. Below, we address each of your points in detail. Please see the General Response for additional information and experiments that clarify several aspects.
> - [Additional models] We share your concern regarding the relevance to state-of-the-art models. To address this, we conducted new experiments using Llama-3.2-1B (see General Response). The results align with our previous findings, suggesting that the relationship between brain alignment and linguistic competence holds for larger, modern models as well.
> - [Effect size and win rate] Regarding effect size, the updated paper includes new results on BERT (trained using LoRa) showing a more pronounced difference than our initial submission (see General Response). We attribute weaker differences in specific cases (e.g., GPT-2) to the model's limited capacity to decrease brain alignment. Regarding win rates, we use them to standardize diverse evaluation metrics across tasks. To avoid masking marginal gains, we report an "adjusted Win Score" (lines 262-269) that credits a win only when the performance difference is statistically significant across seeds. Consequently, win rates do not sum to one, reflecting ties where significance is not reached.
> - [Improved alignment outside language regions] The models are trained using only language-related regions as reported in Lines 168-169; therefore, the observed effects are due to decreased alignment in those specific regions. It is possible that an area outside the language regions increases its alignment instead of decreasing it.
> - [Related work on training models on brain data]  We updated the paper to include papers that investigates brain tuning (Negi et al., 2025)(Swartz et al., 2019)(Vattikonda et al., 2025) and we already mention (Moussa et al., 2025). In the updated paper, we also include Brain Tuned models (see General Response). There are substantial differences, however, because in our case we fine-tune using not only brain data but also text (preserving LM loss); moreover, we evaluate the models on a much larger set of tasks. Future studies can build upon these methodologies to further investigate fine-tuning with brain data.
>
> We hope that our clarifications have addressed the reviewer’s concerns.

---

> > ### Comment · Reviewer_J54j · 2025-11-25
> >
> > Thank you for addressing my questions and concerns. I think this is an interesting approach and worth publishing. I have increased my score.
> >
> > For the related work, please connect it to broader literature beyond language; eg vision researchers have studied the effect of training on neural data more extensively (in the aligned direction eg https://openreview.net/pdf?id=SMYdcXjJh1q rather than this work's focus on misalignment so it doesn't take away any novelty).

---

### Official Review · Reviewer_SLR2 · 2025-11-01

**Soundness:** 2
**Presentation:** 3
**Contribution:** 3
**Rating:** 4
**Confidence:** 1

**Summary:**

This paper studies a simple but important question. Does the alignment between large language models (LLMs) and brain activity matter, or is it just a coincidence? To answer this, the authors use adversarial training to erase the information in the model that can predict brain activity, without affecting the model’s language ability.
They use BERT-base and GPT-small models on two fMRI datasets. As a control, they train other models in which the brain data is shuffled, to rule out effects from the training itself.
They then evaluate these models on over 200 linguistic tasks. Models where "brain alignment" was erased perform worse on semantics, syntax, and reasoning tasks. The authors conclude that brain-aligned representations are important for high-level language ability in LLMs.

**Strengths:**

1. The main contribution is proposing a new paradigm beyond simple correlation studies. The paper creates “brain de-aligned” models through adversarial training, with “brain retention” models as controls. This is a well-designed intervention experiment, providing a new tool for studying the function of representations.
2. Using the Holmes benchmark to evaluate over 200 fine-grained linguistic tasks is a major strength. It allows the authors to see how de-alignment affects various aspects of language, not just a single task.

**Weaknesses:**

1. The key findings are only based on bert-base and gpt-small. In 2025, using these models is not enough to study emergent properties like brain alignment in LLMs. Many key language abilities and representations only show up in much larger models (for example, over 7B parameters). Do these findings hold for modern models like Llama or Qwen?
2.  On the Harry Potter dataset, the difference in GPT-2 model performance is not statistically significant (p=0.055). On the Moth Radio Hour dataset, the authors say the results are inconsistent because the “brain removal” method was weak. Appendix Figure 18 shows that the method only works for 3 out of 6 subjects. This inconsistency (good results for BERT, but not for GPT-2) weakens the main message. The paper does not explore why such architecture differences (encoder vs decoder) produce such different results.

In addition, there is a '??' at line 337, which looks unprofessional in a formal submission.

**Questions:**

see weakness.

---

> ### Author Response · Authors · 2025-11-23
>
> We thank the reviewer for their valuable feedback. In response, we have clarified key points, added new experiments that have strengthened our analysis, and updated the paper accordingly. We address each concern below.
> - [Additional and newer model] As detailed in our General Response, we extending our analysis to Llama-3.2-1B. We found that our results are consistent even with this more modern architecture, confirming that the observed effects of brain alignment generalize beyond BERT and GPT-2.
> - [Different Results for GPT-2] The additional experiments on Llama mentioned above do not support the hypothesis that these discrepancies are due to the different architecture (i.e., decoder-only). Investigating whether this persists in other models of the GPT-2 family is indeed an interesting next step. Additionally, in the General Response, we discuss additional experiments on Brain Tuning (i.e., improving brain alignment instead of decreasing it). Those results point in a similar direction across all models; it seems that for GPT-2, while it was difficult to remove brain alignment, it was feasible to increase it, which has downstream effects on linguistic competence.
>
> We hope that the clarifications we provided have satisfied the reviewer, and we look forward to the opportunity to address any outstanding concerns.

---

### Author Response · Authors · 2025-11-23

We thank all reviewers for their suggestions and believe that addressing their feedback will further increase the impact of our work. We respond to each reviewer individually below, and here we address important concerns that were shared among multiple reviewers or that we think are particularly important.
- [Results with newer language model] Reviewers SLR2, J54j, 7k7B and Fhx9 asked for an analysis using a newer language model. We have now trained both Brain Misaligned and Brain Preserving models using Llama-3.2-1B, for both the Harry Potter and Moth Radio Hour datasets. We chose this model because, as a distilled version of the larger Llama-3.1 family, it offers a representative “modern” architecture (unlike BERT/GPT-2) while maintaining the computational feasibility required to perform the extensive experiments. The results are consistent with the previous two models (BERT and GPT-2) and reveal an increased performance of the Brain Preserving model on linguistic competence with respect to the Brain Misaligned model, indicating that when brain alignment is decreased, it impairs downstream linguistic competence. We have updated the paper with these results in Appendix H and I as well as in the Methodology and Results sections.
- [Results from Brain Tuned and Pretrained models] Reviewer 7k7B proposed strengthening our conclusions by investigating a complementary hypothesis: if decreasing alignment impairs competence, does increasing it lead to performance gains? We thought that this hypothesis could also be relevant for other reviewers who posed questions about the magnitude of the effects. For each of the 6 language model and dataset combinations (3 LMs x 2 brain datasets), we trained a Brain Tuned model, which was fine-tuned using the same procedure as the Brain Misaligned model (i.e. with two heads, one for language modeling and the other for brain mapping), but we removed the gradient reversal layer. We compared this model to the Brain Preserving model and showed that in every model and dataset combination, the Brain Tuned model outperforms the Brain Preserving model, and the difference is statistically significant in the Syntax and Semantics subfields. Additionally, we compared the Brain Tuned model with the Pretrained model. We show that in the majority of model-dataset combinations, the Brain Tuned model outperforms the Pretrained model, in particular for the Semantics and Syntax subfields.
- [Summary results] Following the suggestion from Reviewer 7k7B, we now report averaged results across models and datasets in the main paper and moved the analysis of specific models to the Appendix. This allows for an easier assessment of results that generalize across model and dataset combinations. As already reported above and in the previous version of the paper, overall the Brain Preserving Model outperforms the Brain Misaligned model, in particular for syntax and semantics, with unique differences across model and dataset combinations. In particular statistical tests on each individual model and dataset combinations reveal a marked difference in performance between the Brain Misaligned and Brain Preserving on the overall linguistic competence. For the BERT-based and Llama-based models the difference is significant, although for GPT2-based models on the Harry Potter dataset the difference does not reach conventional statistical significance (p-value $= 0.055$), the trend mirrors the effect observed in the BERT-based models. For the GPT2-based models on the Moth Radio Hour dataset, results are not consistent due to the weaker effect of brain removal.
- [Updated results for BERT] Because we are now reporting averages across models and datasets, we now unify the training settings for all language models using LoRA. The original results used LoRA fine-tuning for GPT-2 but full-finetuning for BERT. We have updated all results in the paper to use LoRA fine-tuning. In this setting the Brain Preserving model significantly outperforms the Brain Misaligned across all tasks, across all linguistic subfields and across the majority of linguistic phenomena.

We hope to have addressed the reviewers' major concerns and that the additional experiments have strengthened our work. We look forward to discussing with reviewers if there are remaining concerns.

---

### Author Response · Authors · 2025-12-02

We thank the reviewers for their constructive feedback. Due to the suspension of further comments on November 28th, Reviewers SLR2 and Fhx9 did not have the opportunity to acknowledge our latest responses and results.

However, it is important to highlight that the primary concerns raised by these reviewers overlap significantly with those raised by Reviewers 7k7B and J54j, who subsequently raised their scores after reviewing the new experiments.
Specifically:
- Regarding the request for Additional Models and Discrepancies in GPT-2 Results (raised by SLR2 and Fhx9): The concern regarding missing additional models was shared by Reviewers 7k7B and J54j. We included results for Llama-3.2-1B, which demonstrated that the findings generalize to modern architectures. The concern regarding GPT-2 results was addressed by the introduction of Llama-3.2-1B and Brain Tuned models, revealing that the results follow a consistent trend across all models. Reviewers 7k7B and J54j acknowledged these improvements.
- Regarding Additional Datasets and Effect Sizes (raised by Fhx9): These points align with feedback from Reviewers 7k7B and J54j, respectively. The methodological justification for dataset selection was accepted in the discussion with Reviewer 7k7B, while the clarification regarding effect sizes resolved the concerns shared by Reviewer J54j.

We hope that the resolution of these shared concerns, validated by the reviewers who were able to engage with the new results, will be taken into full account for the final evaluation of this work.

---

### Meta-Review · Area_Chair_JC7W · 2025-12-12

**Summary:**

This study investigates the practical consequences of LLM-brain (mis)alignment for task performance. When BERT-base and GPT-2-small are optimized to *not* predict fMRI signals, language modeling loss does not increase, but performance on semantics and syntactic tasks decreases. It is argued that this yields causal evidence of brain alignment directly supporting linguistic competence.

Reviewers agreed that treating brain alignment as a property that can directly be intervened upon is novel and interesting. My primary concern is that the experiment does not yield direct causal evidence that brain alignment is what's driving these factors; rather, I think it is more likely that there are many confounds, such as syntax-related representations, that drive both brain alignment and task performance. Thus, running anti-alignment should reduce performance, as it's essentially equivalent to telling the model to explicitly unlearn anything grounded in human cognition. This includes a lot of abilities, so the main finding could be viewed as somewhat tautological. That said, the control experiments are quite good, and do mitigate this concern to a significant degree. Other reviewer concerns include the small effect sizes and outdated models. I think these are related: there could be more brain alignment to destroy if the models were better in the first place. This has been partially addressed via new experiments with Llama-3.2-1B, but further experiments with more recent models could yield stronger findings.

Overall, reviewers were enthusiastic about the general direction, but seemed to remain somewhat skeptical about the experimental setup. Given the clarifications during the discussion period, I believe this work could be of interest and inspiration to those at ICLR, and the experimental flaws have been shown to be more minor than reviewers initially sensed.

**Reviewer Concerns:**

A shared concern is that the models, BERT and GPT-2, are too outdated and small. I am generally unsympathetic to this type of concern in analysis studies, as reviewers tend not to state why this is a concern, such that it feels like a request for more models for their own sake. However, in this study, it may actually affect the takeaways: stronger models would probably have more interesting brain-aligned features to destroy, and this could yield stronger effect sizes. The authors are on the right track with the addition of a Llama experiment, but scaling up further would be beneficial.

Another shared concern is the small effect sizes. This is likely related to the previous point. The authors have run additional experiments during the discussion period that have yielded stronger effect sizes, but the original results are still difficult to square with the main claims.

Finally, a concern from Reviewer 7k7B is that the analyses do not reveal the direct causal effect of alignment, but rather anything that could be correlated with fMRI signals. The control mentioned in the discussion does mitigate this concern to a significant degree. I also think that the authors have gotten closer to showing the effect of this variable than past work. Thus, while not perfectly addressed, I do still think this paper represents an advancement over prior work in measuring the influence of this variable.

Other concerns relating to controls and clarifications have been well-addressed, in my opinion.

**Reviewer Scores:**

Reviewers J54j has stated they had improved their already-positive score. Reviewer 7k7B said that they had adjusted their scores, but didn't say in which direction; I will assume that this was in a positive direction, given my positive assessment of the responses.

I sense that SLR2 would not have increased their scores, as the new results partially but don't completely address their concerns relating to model capabilities: the new Llama model is newer, but is still relatively small and less capable than state-of-the-art open-weights models. While the new experiments have increased the effect size, it is still not entirely clear why the effect sizes were originally so small. Reviewer Fhx9 may have slightly increased their score, given that the response addressed many but not all of the listed concerns.

---

### Decision · Program_Chairs · 2026-01-26

Accept (Poster)